# No Regrets: Investigating and Improving Regret Approximations for Curriculum Discovery

**Alex Rutherford**[*]   **Michael Beukman**[*]
**Timon Willi**   **Bruno Lacerda**   **Nick Hawes**   **Jakob Foerster**

University of Oxford

## Abstract

What data or environments to use for training to improve downstream performance is a longstanding and very topical question in reinforcement learning. In particular, Unsupervised Environment Design (UED) methods have gained recent attention as their adaptive curricula promise to enable agents to be robust to in- and out-of-distribution tasks. This work investigates how existing UED methods select training environments, focusing on task prioritisation metrics. Surprisingly, despite methods aiming to maximise regret in theory, the practical approximations do not correlate with regret but with success rate. As a result, a significant portion of an agent's experience comes from environments it has already mastered, offering little to no contribution toward enhancing its abilities. Put differently, current methods fail to predict intuitive measures of "learnability." Specifically, they are unable to consistently identify those scenarios that the agent can sometimes solve, but not always. Based on our analysis, we develop a method that directly trains on scenarios with high learnability. This simple and intuitive approach outperforms existing UED methods in several binary-outcome environments, including the standard domain of Minigrid and a novel setting closely inspired by a real-world robotics problem. We further introduce a new adversarial evaluation procedure for directly measuring robustness, closely mirroring the conditional value at risk (CVaR). We open-source all our code and present visualisations of final policies here: https://github.com/amacrutherford/sampling-for-learnability.

## 1   Introduction

Curriculum discovery—automatically generating environments for reinforcement learning (RL) agents to train on—remains a longstanding and active area of research [1, 2]. Automated curriculum learning (ACL) methods offer the potential to generate diverse environments, leading to the development of more general and robust agents. Recently, a class of methods under the umbrella of Unsupervised Environment Design (UED) has gained popularity, owing to their theoretical guarantees of robustness and empirical improvements in out-of-distribution generalisation [3–5].

Currently, the most popular and empirically successful subfield of UED develops methods that aim to maximise *regret*—the difference in performance between an optimal agent and the current agent [3–8]. However, computing regret is intractable in all but the simplest tasks, forcing practical methods to instead approximate it [4, 9]. While the prevailing assumption has been that these approximations are faithful, we investigate this further and find that this is not the case: specifically, these scoring functions correlate with *success rate* rather than regret. This means that these methods tend to prioritise tasks that the agent can already solve, leading to much of the collected experience not contributing to learning an improved policy.

---

[*]Equal Contribution. Correspondence to arutherford@robots.ox.ac.uk.

38th Conference on Neural Information Processing Systems (NeurIPS 2024).

While we find that regret performs well in settings where we can compute it—confirming that the underlying theory is sound—we show that the common approximations are unreliable. Therefore, we focus on a different scoring mechanism which instead prioritises levels that provide a clear learning signal to the agent. More specifically, these levels are those where the agent's success rate is neither 100% nor 0%, i.e., levels that are neither too difficult nor too easy [1, 10].

Using this scoring function, we develop *Sampling For Learnability* (SFL), a method that estimates learnability by rolling out the current policy on randomly sampled levels and selecting those that the agent solves sometimes, but not always. We find that this simple and intuitive approach outperforms DR [11], Prioritised Level Replay [4, 9] and ACCEL [5] on four challenging environments, including our novel single- and multi-agent robotic navigation domain, Xland-Minigrid [12] and Minigrid [13].

To truly put our method to the test, we develop a new, more rigorous robustness evaluation protocol for ACL, and demonstrate that SFL significantly outperforms all other methods. Rather than evaluating on a set of arbitrary hand-designed environment configurations, our protocol computes a *risk* metric on the performance of the method, by evaluating its performance in the worst $\alpha\%$ of a newly sampled set of environments.

Our contributions are as follows:

1. We illustrate the inefficacy of current UED methods on several domains, including a novel robot navigation environment.

2. We identify the reason for this observation: current UED regret approximations are flawed.

3. We present *Sampling For Learnability* (SFL), a simple algorithm that trains on environment configurations that have a positive, but not perfect, solve rate, and show that it significantly outperforms current UED approaches on four domains.

4. We introduce a new evaluation protocol: the conditional value at risk (CVaR) of success of a trained agent on a set of sampled levels. This metric specifically measures the risk of poor generalisation, quantifying robustness in the ACL setting.

## 2 Background

### 2.1 Reinforcement Learning & UPOMDPs

We model the reinforcement learning problem as an underspecified partially observable Markov decision process (UPOMDP) [3], denoted by $\mathcal{M} = \langle A, O, \Theta, S, \mathcal{T}, \mathcal{I}, \mathcal{R}, \gamma \rangle$. Here, $A$, $S$, and $O$ represent the action, state, and observation spaces, respectively. The agent receives an observation $o$ (without directly knowing the true state $s$) and selects an action, which results in a transition to a new state, a new observation, and an associated reward. $\Theta$ represents the set of possible parameters, where each $\theta \in \Theta$ defines a specific level. Each $\theta$ corresponds to a particular instantiation of the POMDP, with an associated transition function $P_\theta : S \times A \to \Delta(S)$ and an observation function $\mathcal{I}_\theta : S \to O$.

In a multi-agent setting, $n$ agents make decisions simultaneously. At each step, agent $i$ chooses an action $a_i$, forming a joint action $a = \{a_1, \ldots, a_n\}$ that transitions the environment according to $P_\theta$. Each agent then receives a reward based on the reward function $\mathcal{R} : S \to \mathbb{R}$, which may be shared among all agents or be agent-specific.

### 2.2 Unsupervised Environment Design (UED)

UED is an autocurricula paradigm that frames curriculum design as a two-player zero-sum game between a level-generating adversary and an agent. The agent seeks to maximise its expected return in the standard RL manner, while the adversary can pursue various objectives. Domain Randomisation (DR) fits within this framework by assigning a constant utility to each level, reducing level generation to mere random sampling [11]. Worst-case methods, on the other hand, incentivise the adversary to minimise the agent's reward, aiming to enhance performance on the most challenging levels [14]. However, this approach often results in the generation of unsolvable levels [3].

Regret-based UED methods offer an alternative by generating levels that maximise the agent's regret. Here, the regret of a policy $\pi$ on a level $\theta$ is defined as the difference between the policy's discounted return on $\theta$ and the optimal return achievable on that level, expressed as $U(\pi_\theta^\star) - U(\pi)$, where $U$

denotes the policy's discounted return and $\pi_\theta^\star$ is the optimal policy on $\theta$. This approach deprioritises unsolvable levels and should, in theory, generate levels on which the agent can continue to improve. Moreover, using regret as the adversary's objective provides additional theoretical benefits. At the convergence of the two-player zero-sum game, the agent's policy enjoys minimax regret robustness, meaning that the maximum regret over the entire level space $\Theta$ is bounded [3].

However, computing regret in complex settings is intractable because it requires access to the optimal policy for each level. As a result, current UED methods rely on heuristic score functions that aim to approximate regret. The two most commonly used heuristic approaches are Positive Value Loss (PVL) and Maximum Monte Carlo (MaxMC).

**Positive Value Loss (PVL).** PVL approximates regret as the average of the value loss across all timesteps where it is positive. When using GAE [15] (as in PPO [16]), PVL can be written as follows:

$$\text{PVL} \doteq \frac{1}{T} \sum_{t=0}^{T} \max \left( \sum_{k=t}^{T} (\gamma\lambda)^{k-t} \delta_k, 0 \right),$$

where $\gamma$ is the discount factor, $\lambda$ is the GAE coefficient and $T$ is the length of the episode. $\delta_t$ is the TD-error at timestep $t$, defined as $\delta_t \doteq R_t + \gamma V(o_{t+1}) - V(o_t)$, with $V$ denoting the agent's value function, corresponding to the discounted return by following $\pi$ from $o_t$.

**Maximum Monte Carlo (MaxMC).** Instead of using the bootstrapped value target, MaxMC instead uses the highest return obtained on a particular level:

$$\text{MaxMC} \doteq \frac{1}{T} \sum_{t=0}^{T} \left( R_{\max} - V(o_t) \right).$$

### 2.2.1 Current UED Methods

Prioritised Level Replay (PLR) [4, 9] involves two key steps: generating random levels and replaying levels from a buffer. Initially, random levels are created, and the agent is evaluated on them, with each level being assigned a score. High-scoring levels are then added to a buffer. Subsequently, levels are sampled from this buffer based on their score and the time elapsed since their last selection, and the agent is trained on these sampled levels. PLR has two variants: standard PLR and Robust PLR. In standard PLR, the agent's policy is updated using rollouts from the randomly generated environments, whereas in Robust PLR, the policy is not updated during this phase. ACCEL [5] extends the PLR framework by incorporating a mechanism that randomly mutates previously high-scoring levels, generating new levels that push the agent to the edge of its capabilities. For additional methods, see Section 8, and for a more detailed introduction to ACL, refer to Appendix E.2.

## 3  `JaxNav`

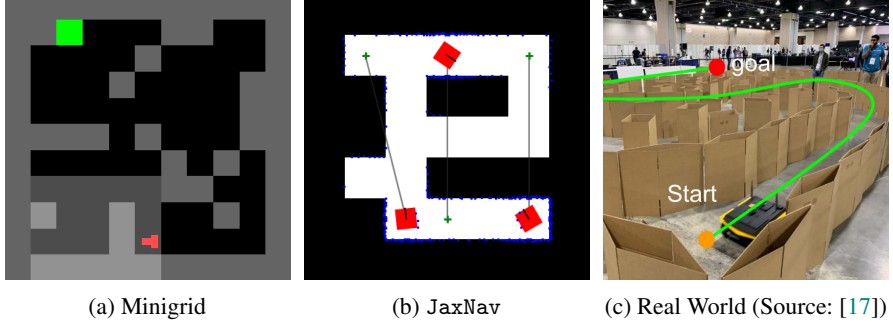

(a) Minigrid      (b) `JaxNav`      (c) Real World (Source: [17])

Figure 1: JaxNAV (b) brings UED, often tested on Minigrid (a), closer to the real world (c)

In this section, we first touch on hardware-accelerated environments, and robotic navigation. We then go on to introduce `JaxNav`, a hardware-accelerated, single and multi-agent robotic navigation environment. While similar on a surface level to Minigrid, `JaxNav` has several additions that make it closer to real-world robotics environments.

## 3.1 Hardware Accelerated Environments

Recently, Bradbury et al. [18] released JAX, a Python numpy-like library that allows computations to run natively on accelerators (such as GPUs and TPUs). This has led to an explosion in reinforcement-learning environments being implemented in JAX, leading to the time it takes to train an RL agent being reduced by hundreds or thousands of times [12, 19–21]. This has enabled researchers to run experiments that used to take weeks in a few hours [22, 23]. One side effect of this, however, is that current UED libraries are written in JAX, meaning they are primarily compatible with the (relatively small) set of JAX environments.

## 3.2 Robot Navigation

Motion planning is a fundamental problem for mobile robotics. The general aim is to find a collision-free path from a starting location to a goal region in two or three dimensions. We focus specifically on the popular setting of 2D navigation problems for differential drive robots using 2D LiDAR readings as the sensory input for their navigation policies. Given range readings, the robot's current velocity and the direction of its goal, the navigation policy must produce velocity commands to move the robot to its goal location while avoiding static and dynamic obstacles.

## 3.3 Environment Description

The observation space of `JaxNav` is highly partially observable and is based on LiDAR readings, depicted by the blue dots in Figure 1b. This is in contrast to Minigrid, which provides a top-down, ego-centric, image-like observation, as shown by the highlighted region in Figure 1a. Additionally, while Minigrid features discrete forward and turn actions, the robots in `JaxNav` operate in continuous space using differential drive dynamics. Similar to Minigrid, agents in `JaxNav` must navigate from a starting location to a goal region, with the goal centre represented by the green cross in Figure 1b.

These design choices make `JaxNav` a close approximation of many real-world robotic navigation tasks [24, 25], including the ICRA BARN Challenge [26], which is depicted in Figure 1c. This challenge, which has run annually since 2022, aims to benchmark single-robot navigation policies in constrained environments for differential drive robots using 2D LiDAR as sensory input. Even with a cell size of 1.0 m, `JaxNav` offers a similar clearance between robots and obstacles as the test maps used in the BARN Challenge, underscoring its relevance not only for evaluating UED methods but also for advancing robotics research. Our environment's full design is outlined in Appendix A.

# 4 Understanding and Improving Level Selection in Goal Directed Domains

In this section, we examine current UED methods, and investigate how they select levels to train on. In particular, we investigate how well currently-used score functions correlate with (a) success rate (i.e., the fraction of times the agent solves the level); and (b) learnability (defined below). We then develop a method which directly samples levels according to their learnability potential, with the following sections detailing our experimental setup and results.

## 4.1 Defining Learnability

Similarly to the *Goals of Intermediate Difficulty* objective proposed by Florensa et al. [1] and the *ProCuRL* curriculum strategy proposed by Tzannetos et al. [10], we desire agents to learn on levels that they can solve sometimes but have not yet mastered. Such levels hold the greatest source of possible improvement for an agent's policy and so a successful autocurricula method must be able to find these. Indeed, given a success rate (i.e., the fraction of times the agent solves the level) of $p$ on a given level, we follow Tzannetos et al. [10] and define learnability to be $p \cdot (1 - p)$. In a goal-based setting where there is only a nonzero reward for reaching the goal, we justify this definition as follows:

1. $p$ represents how likely the agent is to obtain positive learning experiences from a level, while $1 - p$ represents the maximum potential improvement the agent can make on that level. Multiplying these yields (probability of improvement) · (improvement potential), i.e., expected improvement.

2. Tzannetos et al. [10] derive this definition for two specific, simple, learning settings and show that at each training step, selecting for the highest learnability is equivalent to greedily optimising the agent's expected improvement in its training objective.

3. $p \cdot (1 - p)$ can also be seen as the variance of a Bernoulli distribution with parameter $p$, i.e., how inconsistent the agent's performance is.

## 4.2 Analysing Regret Approximations used by UED Methods

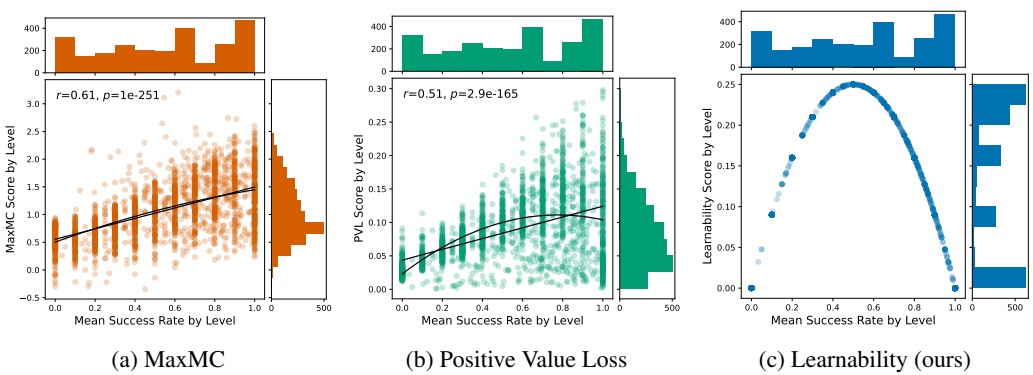

(a) MaxMC         (b) Positive Value Loss         (c) Learnability (ours)

Figure 2: Our analysis of UED score functions shows that they are not predictive of "learnability".

Having defined learnability, we now turn our attention to the current UED score functions. As demonstrated in Section 7, the latest state-of-the-art UED methods fail to outperform Domain Randomisation (DR) in the multi-agent `JaxNav` environment. To highlight the limitations of these approaches, we examine whether their score functions can reliably identify the frontier of learning, i.e., levels that agents can only sometimes solve.

We focus on the single-agent version of `JaxNav` and conduct rollouts using the top-performing seed for PLR-MaxMC on randomly sampled levels over 5000 timesteps. From these rollouts, we compile a set of 2500 levels, evenly distributed into 10 bins based on mean success rate values ranging from 0 to 1. We then perform additional rollouts on this collected set, running for 512 environment timesteps (the same number as used during training) across 10 parallel workers, and average the results. In Figure 2, we plot the mean MaxMC, PVL and Learnability scores against the mean success rate for each level. We additionally report the Pearson correlation coefficient, $r$, and $p$-value for the linear relationship between the success rate and the regret score.

Our analysis reveals no correlation between MaxMC and learnability, and MaxMC instead shows a slight correlation with success rate. While PVL has a weak correlation with learnability, the high variance causes already-solved maps to be prioritised alongside those with high learnability. These plots contrast heavily with that of our learnability metric, which directly prioritises levels with the greatest expected improvement. We hypothesise that the root cause of this issue is the agent's poor value estimation. In a highly partially observable environment, the agent struggles to accurately estimate the value of a state, leading to noisy MaxMC and PVL scores, which in turn hinder UED methods from effectively identifying the learning frontier. Given that reward is strongly correlated with success rate, these findings also apply when comparing scores against reward, as detailed in Appendix F.

## 4.3 Sampling For Learnability: Our Simple and Intuitive Fix

Following on from our analysis, we now present *Sampling For Learnability* (SFL), a simple approach that directly chooses levels that optimise learnability. Our approach maintains a buffer of levels with high learnability and trains on a set of levels drawn from this buffer alongside randomly generated levels. Algorithm 1 outlines the overall approach for SFL and illustrates the relative simplicity of our method compared to SoTA UED approaches. The policy's weights $\phi$ are updated using any RL algorithm; we use PPO [16] for all of our experiments. Meanwhile, our method for collecting learnable levels is detailed in Algorithm 2. We find that the default values of $T = 50$, $\rho = 0.5$, $N_L = 256$, $L = 2000$, $N = 5000$ and $K = 1000$ work well across domains. However, the

per-environment hyperparameters we use are listed in Appendix C, and Appendix I contains plots showing the effect of changing each of these hyperparameters.

---

**Algorithm 1** Sampling For Learnability

> **Initialize:** policy $\pi_\phi$, level buffer $\mathcal{D}$
> **while** not converged **do**
>     $\mathcal{D} \leftarrow$ **collect_learnable_levels**$(\pi_\phi)$ Using Alg. 2
>     **for** $t = 1, \ldots, T$ **do**
>         $\mathcal{D}_t \leftarrow \rho \cdot N_L$ levels sampled uniformly from $\mathcal{D}$
>         $\mathcal{D}_t \leftarrow \mathcal{D}_t \cup (1-\rho) \cdot N_L$ randomly generated levels
>         Collect $\pi$'s trajectory on $\mathcal{D}_t$ and update $\phi$
>     **end for**
> **end while**

**Algorithm 2** Collect learnable levels

> **Input:** policy $\pi$
> $\mathcal{B} \leftarrow N$ random levels
> Rollout $\pi$ for $L$ steps for all $\theta \in \mathcal{B}$
>
> $p \leftarrow$ success rate for each rollout
> `Learnability` $\leftarrow p \cdot (1 - p)$
> **return** Top $K$ levels in $\mathcal{B}$ ranked by `Learnability`

---

While a key limitation of this approach is the additional timesteps required to form the learnability buffer, we find that due to the speed of forward rollouts on JAX-based environments, this does not dramatically increase our overall compute time, see Appendix H for a more detailed discussion.

## 5 Experimental Setup

We now outline the domains used along with our adversarial evaluation protocol. Rather than taking the common approach of reporting average performance on a set of hand designed levels (which by their very nature are arbitrary), we sample a large set of levels and examine each method's performance on their worst-case levels. This directly targets the tails of the level distribution and as such is a superior measure of robustness. We further report the comparative performance of methods on the sampled set to determine the degree to which one method dominates another.

We use four domains for our experiments, `JaxNav` in single-agent mode, `JaxNav` in multi-agent mode, the common UED domain Minigrid [13] and XLand-Minigrid [12]. See Appendix B for more details about the environments. We use 10 seeds for Minigrid and single-agent `JaxNav`, and 5 seeds for multi-agent `JaxNav` and XLand-Minigrid. In all of our plots, we report mean and standard error.

Since SFL performs more environment rollouts, we perform fewer PPO updates in single-agent `JaxNav` and XLand-Minigrid to ensure that SFL uses as much compute time as ACCEL. In Minigrid, the additional environment interactions take a negligible amount of time, so we run the same number of PPO updates for all methods. For multi-agent `JaxNav`, we compare each method using the same number of PPO updates. See Appendix H for more information on the relative speed of each method, and how many updates we run for each method. Generally, the additional SFL rollouts take much less time than the updates themselves, due to the massive parallelisation afforded by hardware-accelerated environments. Additionally, recent work suggests that world-models could also allow more samples to be taken than the base environment allows [27–30], highlighting the future potential of this approach.

We compare against several state-of-the-art UED methods as baselines, implemented with `JaxUED` [22]. We use **ACCEL**, with the MaxMC score function, where the agent trains on randomly generated, mutated and curated levels. We also include a "robust" version [4], where no gradient updates are performed on the former two sets of levels. This uses three times as many environment interactions and is roughly twice as slow as SFL for single-agent `JaxNav`. We use **PLR** with both the PVL and MaxMC score functions. We also include a robust version of PLR which only performs gradient updates on the curated levels; this uses twice as many environment interactions and is 80% slower than SFL on single-agent `JaxNav`. We also use Domain Randomisation (**DR**), which trains only on randomly generated levels, with no curation or prioritisation.

## 6 A Risk-Based Evaluation Protocol

The standard approach to evaluating UED agents is to test them on a set of hand-designed holdout levels [3–5]. Whilst this evaluation approach illustrates the performance of agents on human-relevant tasks, we believe it has several limitations. First, the hand-designed levels are arbitrary, so performance on them is not representative of general performance; second, it does not test a central claim of UED: that it trains agents which are robust to worst-case (yet solvable) environments. To address this, we

propose a novel evaluation protocol that rectifies both of these problems: measuring the conditional value at risk (CVaR) of the success of the trained agent on a set of sampled levels.

To calculate the CVaR we sample $N$ (10, 000 in practice) random, but solvable,[2] levels and rollout the agent policy for 10 episodes on each level. We then find the $\alpha\%$ of levels on which the agent performs worst. To mitigate bias, we rollout the agent again on these levels, and report the average success rate on this $\alpha\%$ subset, i.e., the CVaR at level $\alpha$. This metric directly quantifies the performance of a training method *on its own worst-case levels*, which measures its ability to produce robust agents. We perform this computation for each seed, and report the mean and standard error over seeds, at various $\alpha$ levels. We further use the $N$ sampled levels to calculate the following metrics.

**Mean Success Rate** We average the success of each method over all $N$ levels and then report this as the mean success rate and its standard error over multiple independent seeds. Due to space requirements, these results are included in Appendix G.

**Domination Comparison** To identify the degree to which one method dominates another, we obtain the average solve rate of each method (averaging over seeds) per environment. We then plot a heatmap, where cell $(x, y)$ contains the number of levels method $A$ solves $x\%$ of the time while method $B$ solves them $y\%$ of the time. This metric measures how many environments one method strictly solves more often than another.

# 7 Results

## 7.1 Single-Agent `JaxNav`

Figure 3a shows the CVaR results on single-agent `JaxNav`. We find that optimising for learnability— as our method does—results in superior robustness over a wide range of $\alpha$ values, despite all methods performing similarly with $\alpha = 100\%$ (which amounts to expected success rate over the entire distribution). In this plot, we also plot the results of an oracle method named *Perfect Regret*. This uses the same procedure as SFL but with the score function: $1 - p(\text{success})$. Importantly (and different to all other methods), this method only samples solvable levels, so this metric corresponds closely to regret. While not shown here, using the same metric with unrestricted level sampling—which is a more realistic setting—performs poorly due to it prioritising unsolvable levels. In Figure 4, we perform pairwise comparisons of each baseline against our approach. We find that there are a large number of environments that all methods solve (the bright top-right corner). However, the bottom-right is generally brighter than the top-left, indicating that SFL performs better in general. Overall, SFL's superiority, and Perfect Regret's strong performance, indicates that the flawed approximations of regret are responsible for UED's lack of performance. We provide further evidence for this claim in Appendix I.2, where we use learnability as a score function within PLR and ACCEL.

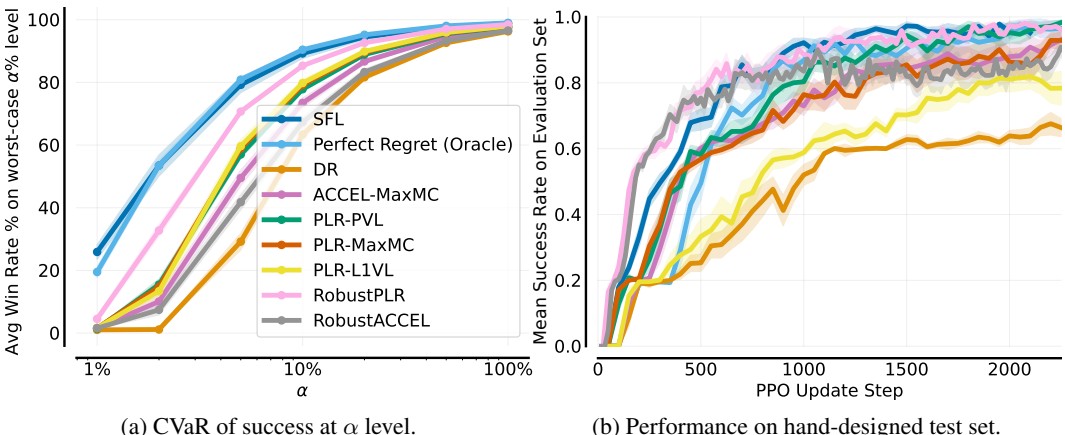

(a) CVaR of success at $\alpha$ level.          (b) Performance on hand-designed test set.

Figure 3: Single-agent `JaxNav` performance on (a) $\alpha$-worst levels and (b) a challenging hand-designed test set. Only *Perfect (Oracle) Regret* matches SFL across both metrics.

---

[2]Solvable means that the goal state can be reached in a particular level, i.e., it is not impossible to complete.



Figure 4: Single-agent `JaxNav` comparison results. For each figure, cell $(x, y)$ indicates how many environments have method $X$ solving them $x\%$ of the time and method $Y$ solving them $y\%$ of the time. The density below the diagonal indicates that SFL is more robust than DR, ACCEL and PLR.

## 7.2 Multi-Agent `JaxNav`

Figures 5 and 6 illustrate the performance of all methods on multi-agent `JaxNav` throughout training. We train with 4 agents and report performance over both a hand designed test set and a randomly sampled set of 100 maps. The levels used in the hand designed set are given in Appendix D and feature cases with 1, 2, 4 and 10 agents. The levels in the sampled set all feature 4 agents and solvability is checked for each agent's individual path. As we train with IPPO, regret scores are calculated on a per-agent basis and the score of a level is computed as the mean across individual agent scores. For $n$ agents, learnability is computed as $\sum_{i=1}^{n}(p_i \cdot (1 - p_i))$, where $p_i$ is the success rate for agent $i$ on on a given level. We find that `JaxNav` significantly outperforms all UED methods.

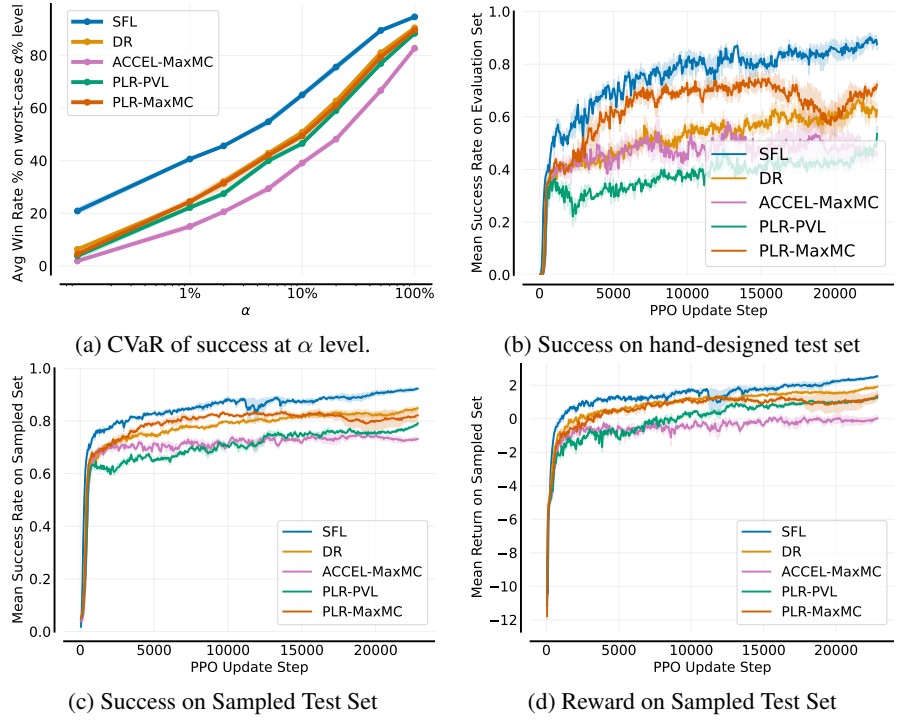

(a) CVaR of success at $\alpha$ level.

(b) Success on hand-designed test set

(c) Success on Sampled Test Set

(d) Reward on Sampled Test Set

Figure 5: Performance of Multi-Agent Policies over 5 seeds. SFL outperforms all UED baselines in each of these. We do not include *oracle regret*, since it cannot be easily measured in this setting.

## 7.3 Minigrid

We next move on to the standard domain of Minigrid (see Figure 7). Here we find that most methods perform similarly on the hand-designed test set; however, SFL significantly outperforms all other methods on the adversarial evaluation, indicating it results in more robust policies.

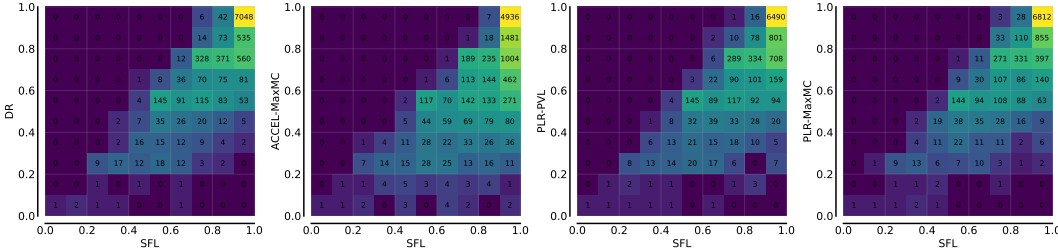

Figure 6: Multi-agent heatmap results. The bright areas toward the right of the plots indicate that SFL outperforms the baselines we compare against.

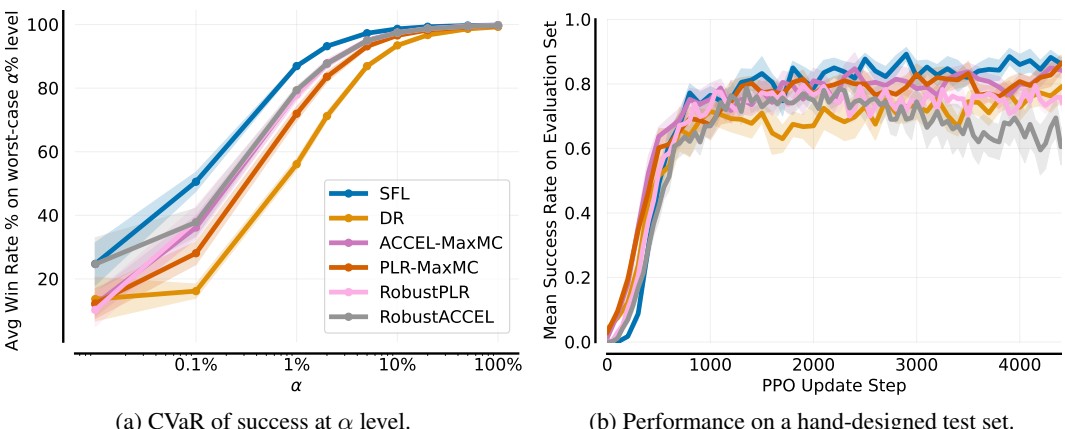

(a) CVaR of success at $\alpha$ level.

(b) Performance on a hand-designed test set.

Figure 7: Minigrid performance on (a) $\alpha$ worst-case and (b) holdout levels. SFL is more robust than the baselines on worst-case levels.

## 7.4 XLand-Minigrid

Our final evaluation domain is XLand-Minigrid's [12] meta-RL task using their `high-3m` benchmark. We report performance using our CVaR evaluation procedure and, in line with [12], as the mean return on an evaluation set during training. Our results are presented in Figure 8, with SFL outperforming both PLR and DR. During evaluation each ruleset was rolled out for 10 episodes. Due to the large number of levels being rolled out to fill SFL's buffer, SFL was slower than DR and PLR. As such, we report results for SFL compute-time matched to PLR.

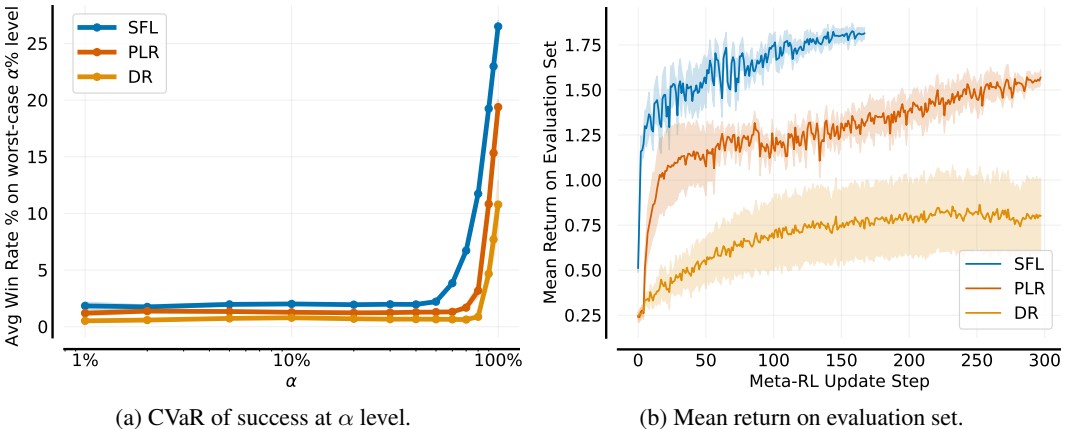

(a) CVaR of success at $\alpha$ level.

(b) Mean return on evaluation set.

Figure 8: XLand-Minigrid performance over five seeds on (a) $\alpha$ worst-case and (b) the evaluation set. SFL outperforms both PLR and DR.

# 8 Related Work

Unsupervised Environment Design (UED) has emerged as a prominent method in the ACL field, promising robust agent training through adaptive curricula. Early works focused on learning potential, where the improvement in an agent's performance determined the choice of training levels [1, 2, 31, 32]. However, robustness-oriented methods such as adversarial minimax introduced the notion of training on levels that minimise agent performance, though these often resulted in infeasible scenarios offering no learning benefits [3, 14, 33]. Minimax regret (MMR), a more refined robustness approach, alleviates some of these issues by ensuring the chosen levels are learnable [3–5]. However, recent work [8] demonstrated that even true regret does not always correspond to learnability, and this mismatch can lead to stagnation during training. Our work extends this line of research by utilising a scoring mechanism that estimates expected improvement, targeting environments with a positive but not perfect solve rate. Unlike existing MMR methods, our approach directly optimises for learnability, instead of using an imperfect proxy for regret, leading to more effective training on our domains.

Relatedly, Tzannetos et al. [10] introduce *ProCuRL*, which uses a similar learnability score as SFL (and further introduce an approximation to the solve rate $p$ using the agent's value function). However, their problem setting is distinct from ours as they assume only a limited fixed pool of tasks are utilised during training, with the goal of improving an agent's performance over a uniform distribution over this pool. We, instead, consider the standard UED setting where we can sample an effectively unbounded number of tasks from some large distribution $\Theta$, with the goal of achieving an agent that is *robust* to worst-case settings and can generalise to unseen problems. Following on from this, Tzannetos et al. [34] extend *ProCuRL* to the setting where the target distribution of tasks is given, and they take into account both how learnable the selected task is, as well as how correlated it is with learnable tasks from the target distribution. While this approach outperforms the original method, it tackles a different problem to SFL and UED in general, i.e., where the target distribution is known.

Finally, robust RL methods have the goal of improving an agent's robustness to environmental disturbances, and worst-case environment dynamics [35–47]. However, these methods generally consider continuous perturbations instead of a mix of discrete and continuous environment settings. Furthermore, these methods tend to be overly conservative and prioritise unsolvable levels.

# 9 Discussion and Limitations

In this work we only consider deterministic, binary outcome domains and due to the nature of the learnability score, SFL is only applicable to such settings. In other domains, we could potentially reuse the intuition that $p(1-p)$ is the variance of a Bernoulli distribution; in a continuous domain, an analogous metric would be the variance of rewards obtained by playing the same level multiple times. Furthermore, our implementation of SFL is in JAX but the method is general. However, one must take the cost of SFL's additional environment rollouts into account when considering implementing our algorithm; we chose JAX because its speed and parallelisation significantly alleviates this constraint. Next, while most current SoTA UED methods, including SFL, randomly generate and curate levels, this approach may become infeasible when the environment space is vast, as random generation may have a very low likelihood of generating valid levels. Finally, while `JaxNav` does have deterministic dynamics, Fan et al. [48] successfully transferred an RL-based multi-robot navigation policy from a simulator of identical fidelity to the real world, suggesting this should be equally possible with `JaxNav`.

# 10 Conclusion

In this paper, we investigate the scoring functions used by current regret-based UED methods and analyse whether they can accurately approximate regret. We find that this is not the case and that these prioritisation metrics instead correlate with success rate, leading to a large amount of experience not contributing to learning an improved policy. Inspired by this analysis, we develop a method based on an intuitive notion of learnability and find that this improves the robustness of the final policies. We also introduce a new robustness-measuring evaluation protocol, reporting a risk measure on performance over the $\alpha\%$ worst-case (but solvable) levels for each method. We hope that our findings inspire future work on more general, domain-agnostic scoring functions, and we open-source all of our code to facilitate this process. Ultimately, we believe this work is a stepping stone towards bridging the gap between popular testbeds for UED and real-world applications.

## Acknowledgements

This work received funding from the EPSRC Programme Grant "From Sensing to Collaboration" (EP/V000748/1). MB is funded by the Rhodes Trust. JF is partially funded by the UKI grant EP/Y028481/1 (originally selected for funding by the ERC). JF is also supported by the JPMC Research Award and the Amazon Research Award.

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

# Appendix

We structure the appendix as follows. Appendix A includes more details about `JaxNav`, and Appendix B describes the other environments we use. The hyperparameters we use and the hand-designed test sets can be seen in Appendices C and D, respectively. Appendix E discusses multi-robot navigation and automated curriculum learning in more detail.

We next provide more results, including more analysis on the UED score functions in Appendix F, additional general results in Appendix G and compute-time analysis in Appendix H. Finally, we thoroughly ablate SFL in Appendix I.

## A `JaxNav` Specification

The environment is designed as follows, with full parameters listed in Appendix C.

**Observations**  The robot's observation at a given timestep $t$ is $\mathbf{o}_t = [\mathbf{l}_t, \mathbf{d}_t, \mathbf{v}_t]$ containing the current LiDAR range readings ($\mathbf{l}_t$), the direction to the robot's goal ($\mathbf{d}_t$), and the robot's current linear and angular velocities ($\mathbf{v}_t$). The LiDAR range readings $\mathbf{l}_t$ is a vector containing the 100 most recent LiDAR range readings from a 360° arc centred on the robot's forward axis, the LiDAR's max range $D_{\text{lidar}}$ is set to 6 m. Given the robot's current position $\mathbf{p}_t$ and the goal position $\mathbf{g}$, the robot's goal direction $\mathbf{d}_t$ is defined as:

$$\mathbf{d}^t = \begin{cases} \texttt{polar}\,(\mathbf{g} - \mathbf{p}_t) & \text{if } ||\mathbf{g} - \mathbf{p}_t|| \leq D_{\text{lidar}} \\ \texttt{polar}\left(\frac{\mathbf{g}-\mathbf{p}_t}{||\mathbf{g}-\mathbf{p}_t||} \cdot D_{\text{lidar}}\right) & \text{otherwise} \end{cases}, \tag{1}$$

where `polar` converts a Cartesian vector to its polar representation. All observation entries are normalised using their maximum possible values.

**Actions**  The policy selects a two-dimensional continuous action $\mathbf{a}_t = [v_t^x, w_t^z]$ representing a target linear ($v_t^x$) and angular velocity ($w_t^z$). The possible linear and angular velocities are limited to a set range, with actions outside these ranges clipped to be within.

**Dynamics**  The action $\mathbf{a}^t$ is translated into movement in the x-y plane using a differential drive kinematics model [49] which includes limits on linear and angular acceleration.

**Rewards**  Our reward function is inspired by Long et al. [50] and aims to avoid collisions while minimising the expected arrival time. Due to the difficulty of the task we include shaping terms which give a small dense reward at each timestep. The reward $r$ received at timestep $t$ is defined as the sum: $r_t = r_t^g + r_t^c + R_{\text{time}}$, where $r_t^g$ rewards the robot for reaching the goal, $r_t^c$ penalises collisions, and $R_{\text{time}}$ is a small penalty at each timestep equal to $-0.01$. The goal reaching term $r_t^g$ is defined as:

$$r_t^g = \begin{cases} R_{\text{goal}} & \text{if } ||\mathbf{p}^t - \mathbf{g}|| < D_{\text{goal}} \\ w_g(||\mathbf{p}^t - \mathbf{g}|| - ||\mathbf{p}^{t-1} - \mathbf{g}||) & \text{otherwise} \end{cases}, \tag{2}$$

where $R_{\text{goal}} = 4.0$, $D_{\text{goal}} = 0.3$ and $w_g = 0.25$. This term rewards the agent for reaching the goal, and provides a small dense reward if the agent moves closer to the goal. Meanwhile, the collision penalty term $r_c^t$ is defined:

$$r_c^t = \begin{cases} R_{\text{collision}} & \text{if collision} \\ R_{\text{close}} & \text{if } \min(\mathbf{l}_t^u) \leq D_{\text{close}} \\ 0 & \text{otherwise} \end{cases}, \tag{3}$$

where $\mathbf{l}_t^u$ are the un-normalised LiDAR readings at timestep $t$, $R_{\text{collision}} = -4$, $R_{\text{close}} = -0.1$, and $D_{\text{close}} = 0.4$m. This term avoids collisions and provides a small dense penalty when the agent is close to obstacles, this encourages safe behaviour.

**Multi-Agent Reward**  In the multi-agent version of `JaxNav`, the reward for each agent $i$ is defined as $\lambda r_i + (1 - \lambda_i) \sum_j^n r_j$, i.e., it shares its own reward, as well as the team reward. We use $\lambda = 0.5$.

# B  Environment Description

Here we describe the other environments we use, with environment parameters listed in Tables 1 to 3.

Table 1: `JaxNav` Parameters

| Parameter | Value |
|---|---|
| Num Agents | 1 (4 for multi-agent) |
| Square agent width | 0.5 m |
| Grid cell size | 1.0 m |
| **Dynamics** | |
| Goal Radius, $D_{\text{goal}}$ | 0.3 m |
| Min linear velocity | 0.0 m/s |
| Max linear velocity | 1.0 m/s |
| Max linear acceleration | 1.0 m/s$^2$ |
| Min angular velocity | -0.6 rad/s |
| Max angular velocity | 0.6 rad/s |
| Max angular acceleration | 1.0 rad/s$^2$ |
| Timestep length | 0.1 s |
| Number of LiDAR beams | 200 |
| LiDAR range resolution | 0.05 m |
| LiDAR max range, $D_{\text{lidar}}$ | 6 m |
| LiDAR min range | 0 m |
| Map size | $11 \times 11$ m |
| Maximum wall fill % | 60% |
| **Reward Signal** | |
| Goal reaching reward, $R_{\text{goal}}$ | 4.0 |
| Distance change weight, $w_g$ | 0.25 |
| Collision penalty, $R_{\text{collision}}$ | -4.0 |
| LiDAR threshold, $D_{\text{close}}$ | 0.4 m |
| LiDAR penalty, $R_{\text{close}}$ | -0.1 |
| Timestep penalty, $R_{\text{time}}$ | -0.01 |

Table 2: Minigrid Parameters

| Parameter | Value |
|---|---|
| Number of walls | 60 |
| Agent view size | 5 |

Table 3: XLand-Minigrid Parameters

| Parameter | Value |
|---|---|
| Ruleset benchmark | high-3m |
| Environment ID | R4-13x13 |
| Image observations | False |

## B.1  Minigrid

Minigrid is a goal-oriented grid world where a triangle-like agent must navigate a 2D maze. As illustrated in Figure 1a, the agent only observes a small region in front of where it is facing and must explore the world to move to a goal location.

## B.2  XLand-Minigrid

This domain combines an XLand-inspired system of extensible rules and goals with a Minigrid-inspired goal-oriented grid world to create a domain with a diverse distribution of tasks. Each task is specified by a ruleset, which combines rules for environment interactions with a goal, and [12] provide a database of presampled rulesets for use during training. Following [12], we use a 13x13 grid with 4 rooms and sample rulesets from their high diversity benchmark with 3 million unique tasks. As training involves sampling from a database of precomputed rulesets, ACCEL is not applicable. PLR and SFL select rulesets for each meta-RL step to maximise return on a held-out set of evaluation rulesets.

# C  Hyperparameters

Table 4 contains the hyperparameters we use, with their selection process for each domain outlined below. We tuned PPO for DR for each domain and then used these same PPO parameters for all methods, tuning only UED-specific parameters.

Table 4: Learning Hyperparameters.

| Parameter | JaxNav Single-Agent | JaxNav Multi-Agent | Minigrid | XLand |
|---|---|---|---|---|
| **PPO** | | | | |
| Number of Updates | 2250 | 22850 | 4500 | - |
| Number of Meta-Steps | - | - | - | 298 |
| # of PPO Updates per Meta-Step | - | - | - | 128 |
| $\gamma$ | 0.99 | | | |
| $\lambda_{\mathrm{GAE}}$ | 0.95 | | | |
| PPO number of steps | 512 | | 256 | 32 |
| PPO epochs | 4 | | | 1 |
| PPO minibatches per epoch | 4 | | | 16 |
| PPO clip range | 0.04 | | | 0.2 |
| PPO # parallel environments | 256 | | | 8192 |
| Adam learning rate | 2.4e-4 | | | 1e-3 |
| Anneal LR | yes | | | |
| Adam $\epsilon$ | 1e-5 | | | 1e-8 |
| PPO max gradient norm | 0.5 | | | |
| PPO value clipping | yes | | | |
| return normalisation | no | | | |
| value loss coefficient | 0.5 | | | |
| entropy coefficient | 0.0 | | | 0.01 |
| Fully-connected dimension size | 512 | | | [16, 256] |
| Hidden dimension size | 512 | | | 1024 |
| **PLR** | | | | |
| Replay rate, $p$ | 0.5 | | | 0.95 |
| Buffer size, $K$ | 1000 | | 8000 | 40000 |
| Scoring function | MaxMC | | | |
| Prioritisation | Top K | | Rank | Rank |
| Temperature, $\beta$ | - | | 1.0 | 1.0 |
| $k$ | 32 | | - | - |
| staleness coefficient | 0.3 | | | |
| Duplicate check | no | | | |
| **ACCEL** | | | | |
| Number of Edits | 5 | | 20 | - |
| Buffer size, $K$ | 8000 | | | - |
| Prioritisation | Rank | | | - |
| Temperature, $\beta$ | 1.0 | | | - |
| **SFL** | | | | |
| Batch Size $N$ | 5000 | | 25000 | 40000 |
| Rollout Length $L$ | 2000 | | | 5070 |
| Update Period $T$ | 50 | 50 | 100 | 4 |
| Buffer Size $K$ | 100 | 100 | 1000 | 8192 |
| Sample Ratio $\rho$ | 1.0 | 1.0 | 0.5 | 1.0 |

## C.1 `JaxNav`

For PPO, we conducted an extensive sweep on the JaxNav environment ensuring robust DR performance. We only tuned hyperparameters for single-agent `JaxNav`, and used the best hyperparameters for multi-agent `JaxNav`. `JaxNav` hyperparameter searches were done over 3 seeds.

For PLR, we performed a grid search, over replay probabilities $\{0.5, 0.8\}$, buffer capacity $\{1000, 4000, 8000\}$, prioritisation $\{\mathrm{rank}, \mathrm{topk}\}$, temperature $\{0.3, 1.0\}$, and $k$ $\{1, 32, 128\}$. For ACCEL, we searched over the same set and additionally included the number of edits, where we considered values of $\{5, 20, 50\}$.

For SFL, we performed line searches over $N$ of $\{500, 5000, 25000\}$, $L$ $\{1000, 2000, 4000\}$, $K$ of $\{100, 1000, 5000\}$, $T$ of $\{10, 50, 500, 1000, 2000\}$ and $\rho$ of $\{0.25, 0.5, 0.75, 1.0\}$.

Since this is a line search and not a grid search, the total number of tuning runs (and total compute) is less than for PLR and ACCEL (60 runs for SFL vs 90 for PLR and 270 for ACCEL).

## C.2   Minigrid

For Minigrid, our JaxNav PPO parameters performed similarly to those given in the JaxUED implementation but allowed us to use 256 environment rollouts in parallel during training compared to JaxUED's 32. For the UED-specific parameters we used the same sweep settings as for `JaxNav`, again conducting the search over 3 seeds.

## C.3   XLand-Minigrid

For PPO, we used the default parameters provided by [12] and the search for UED parameters was conducted over 1 seed. For PLR, we conducted a grid search over replay probabilities $\{0.5, 0.95\}$, buffer capacity $\{20000, 40000\}$, prioritisation $\{\mathrm{rank}, \mathrm{topk}\}$, temperature $\{0.3, 1.0\}$, score function $\{\mathrm{MaxMC}, \mathrm{PVL}\}$. For SFL, initial experiments illustrated that, due to the number of environment rollouts used to fill the buffer, it was slower than PLR and DR. To use a similar compute budget as PLR, we conducted the sweep over only 70B timesteps. For SFL, we performed a grid search over $N$ of $\{40000, 30000\}$, $L$ $\{5070, 7650\}$, $K$ of $\{8192\}$, $T$ of $\{1, 2, 3, 4\}$ and $\rho$ of $\{0.75, 1.0\}$.

# D  Hand Designed Test Sets

The hand-designed levels used for evaluating policy performance throughout training are illustrated in Figures 9 and 10. The set used for multi-agent policies also includes the first 3 maps in Figure 9. Minigrid's levels are shown in Figure 11 and are the same as those used by JaxUED.

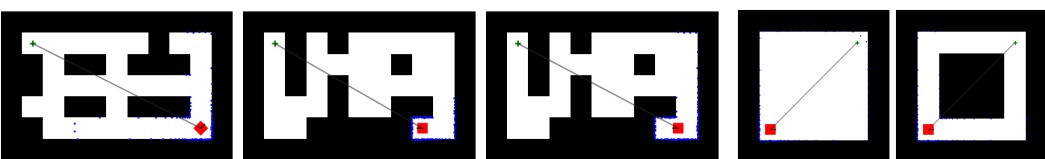

Figure 9: Hand-Designed Test Set for Single Agent JaxNav Policies.

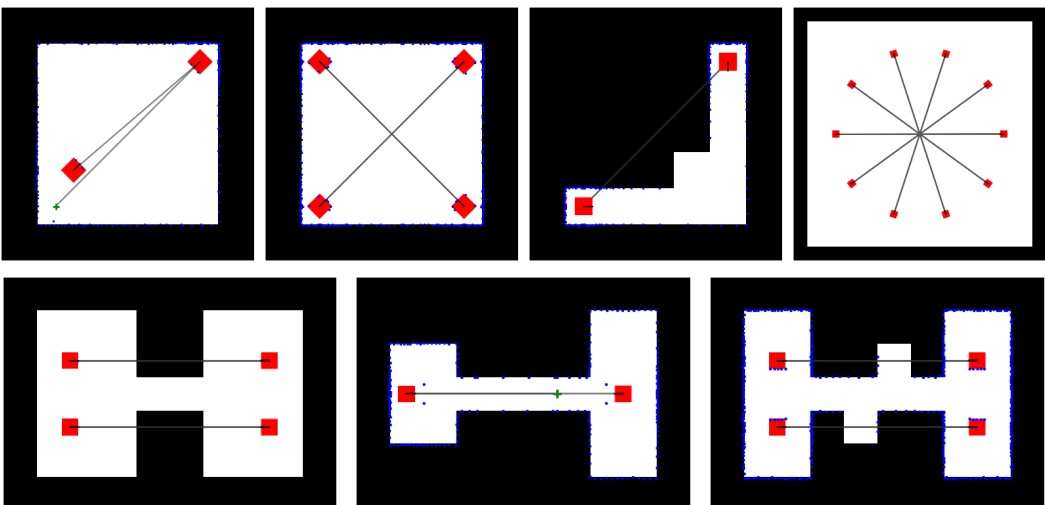

Figure 10: Hand-Designed Test Set for Multi Agent JaxNav Policies.

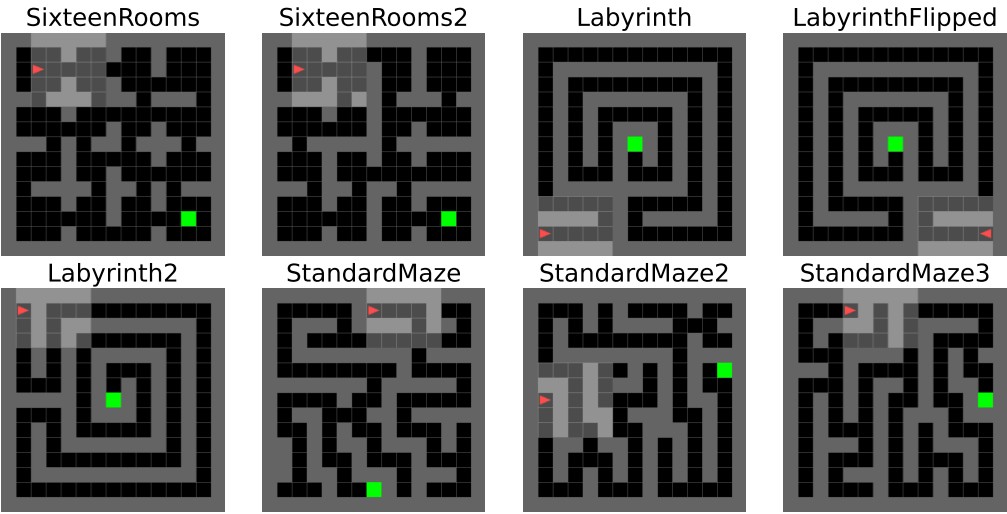

Figure 11: Hand-designed Minigrid Levels [3, 22, 23].

# E    Additional Background Related Work

## E.1    Literature Review of Multi-Robot Navigation

Multi-robot path planning presents unique challenges due to the need for coordination among robots to avoid deadlocks in dynamic environments. Traditional methods often discretise the environment, turning the problem into a multi-agent pathfinding task managed by a central planner [51, 52]. While effective at scale, these approaches rely heavily on communication infrastructure, making them impractical in scenarios with unreliable connectivity or third-party obstacles. In contrast, our method leverages decentralised learning approaches without relying on centralised communication, making it more robust in partially observable and communication-limited settings.

Decentralised approaches like velocity obstacles [53] and ORCA [54] offer a solution by mapping environmental constraints into the robot's velocity space. However, these methods are susceptible to measurement errors and often exhibit short-sighted behavior, limiting their real-world applicability [17]. Our method, by comparison, incorporates a broader evaluation framework that assesses performance in adversarial and challenging environments, ensuring robustness beyond simple dynamic obstacle avoidance.

Recent advancements leverage machine learning to overcome these limitations. CADRL [55] uses RL to address short-sightedness, but its state-based representation limits its adaptability. Our approach, instead, employs lidar-based observations to model a more realistic and complex navigation task, which allows for better generalisation to real-world scenarios.

More sophisticated RL-based approaches, such as those by Long et al. [50] and Tan et al. [56], demonstrate improved performance in open spaces but struggle in constrained settings. Our method specifically addresses this by focusing on environments that are solved intermittently, thereby enhancing the agent's ability to learn in varied and complex settings.

Hybrid methods, combining RL with conventional planning [48], show promise but do not fully address these challenges. In contrast, our method integrates an adaptive curriculum that dynamically adjusts based on the agent's performance, leading to sustained learning improvements even in diverse and adversarial environments.

The design of reward functions is critical in RL-based navigation. Enhancements using velocity obstacles [17, 57] improve performance but still face challenges in real-world transferability. Techniques like perceptual hallucination [58] further enhance robustness by reducing multi-robot planning to static obstacle avoidance, though they typically consider simple scenarios and do not account for dynamic third-party obstacles. Our method, however, introduces a novel evaluation protocol that rigorously tests the robustness of learned policies in a variety of adversarially generated environments, ensuring better real-world applicability.

## E.2    Background on Automated Curriculum Learning

Automated Curriculum Learning (ACL) is a subfield of RL where agents are presented with increasingly challenging tasks that are adapted to the agent's current progress [59, 60]. One common idea idea is to train the agent on tasks that are neither too easy nor too hard, such that it achieves maximum learning potential [1, 61]. Autocurricula methods have various aims, such as improving learning speed on a set of target environments [62] or increasing robustness to unknown environment configurations [3, 4]. Unsupervised environment design (UED) focuses on the latter. One commonality between autocurricula methods is that the environment generator controls aspects of the environment, such as the transition dynamics, state and observation spaces, goals, and so on  [1, 3]. Each of these environment configurations is commonly referred to as a *level* [3].

Methods also differ in how they generate these environment configurations. One class of methods uses generative models, such as Gaussian Mixture Models (GMMs) [2]. While this approach generally makes the problem theoretically tractable, GMMs are limited to continuous-valued parameter settings. More recently, other generative models, such as Variational Autoencoders, have been used [62]. However, these models often require data to train, which may be unavailable or bias the learning process. Other methods use an RL-based level generator, where the generator's objective is based on how the agent performs on the generated level [3]. This approach has been surpassed by the more recent technique of randomly generating and curating levels [4, 5, 9].

# F   Extended analysis of UED Score Functions

An extension of Section 4.2, Figure 12 illustrates the correlation between mean reward and the two most popular regret scores, MaxMC and PVL. These graphs illustrate that the trends seen for success rate also hold for episodic reward, this is expected for our environment as the two are strongly correlated. In Figure 13 we conduct the same analysis for the L1 Loss Score [9], defined as:

$$\text{L1 Loss} \doteq \frac{1}{T} \sum_{t=0}^{T} \left| \sum_{k=t}^{T} (\gamma\lambda)^{k-t} \delta_k \right| .$$

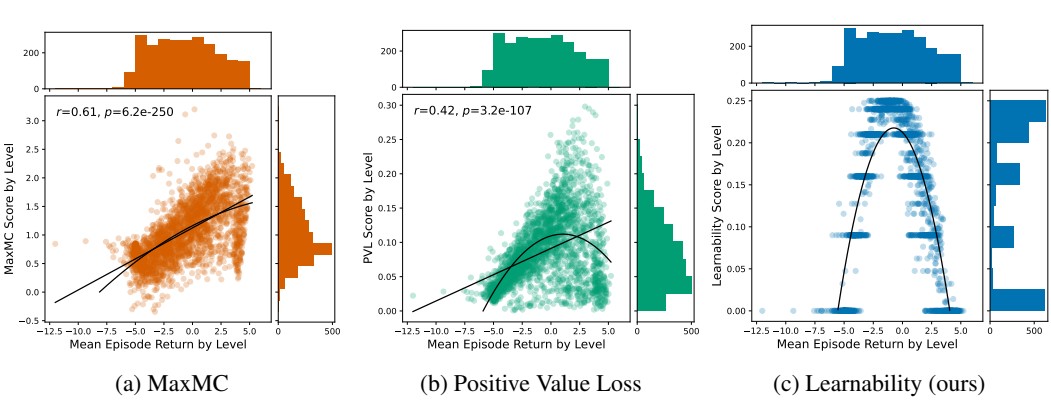

(a) MaxMC  (b) Positive Value Loss  (c) Learnability (ours)

Figure 12: Analysis of UED and Learnability Score Functions Against Reward

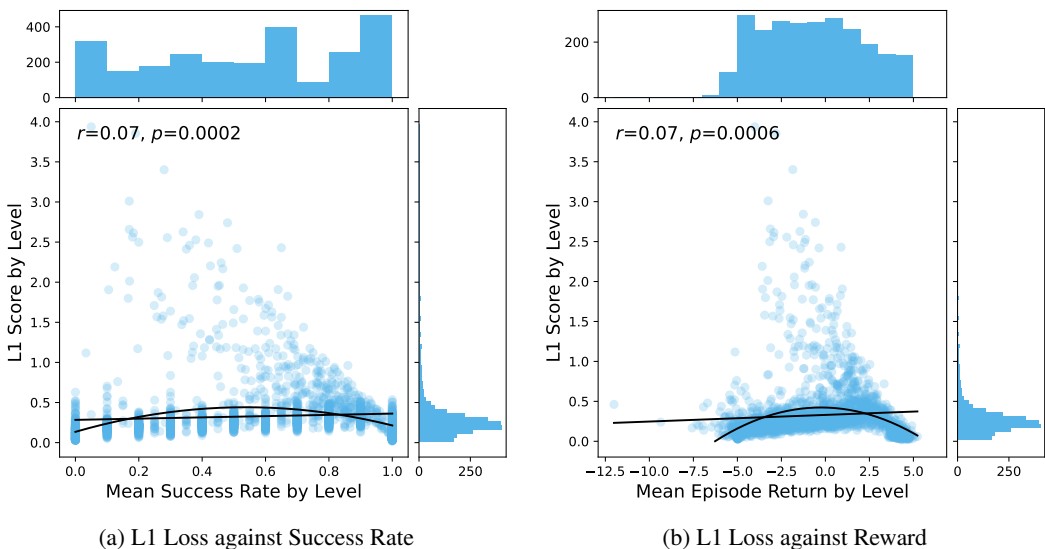

(a) L1 Loss against Success Rate  (b) L1 Loss against Reward

Figure 13: Analysis of L1 Loss score function

# G Additional Results

Figure 14 shows each method's overall solve rate on the set of 10000 sampled solvable levels on a log scale. We find that while all methods solve the vast majority of levels, SFL slightly outperforms all the baseline methods. Figure 15 reports the pairwise comparisons of each base against SFL for Minigrid. While, again, most methods solve most levels, SFL has a slight advantage.

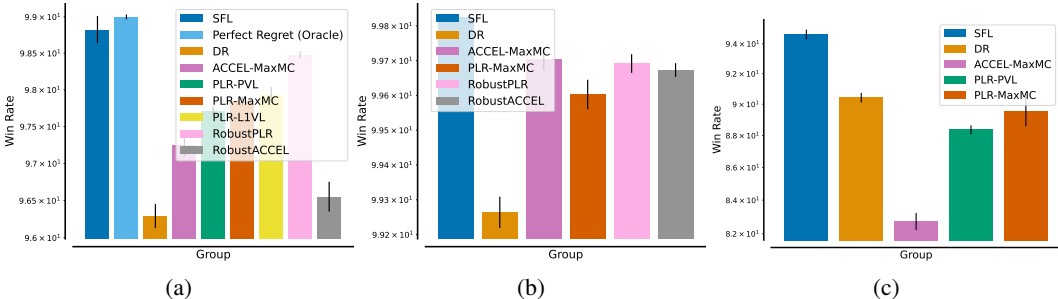

Figure 14: Overall solve rate on 10000 sampled levels on (a) Single-agent `JaxNav` (b) Minigrid and (c) Multi-agent `JaxNav`.



Figure 15: Minigrid results. For each figure, cell $(x, y)$ indicates how many environments have method $X$ solving them $x\%$ of the time and method $Y$ solving them $y$ of the time. In each plot we compare a different baseline to our learnability measure.

## G.1 Episodic Return Plots

Figure 16 shows episodic return and success rate plots for single-agent `JaxNav` and Minigrid. We find that the episodic return is very strongly correlated with the success rate, which is why we primarily show the latter in the main text.

## G.2 Easy Level Analysis

To assess performance on easy levels we have run our evaluation procedure over 10,000 uniformly sampled levels with fewer obstacles than usual. For JaxNav, we used a maximum fill % of $\leq 30\%$, half of the standard 60%. Meanwhile, for Minigrid, we use a maximum number of 30 walls instead of 60. These levels, therefore, are generally easier than the levels we evaluated on in the main paper. Results are reported in Figure 17

On `JaxNav`, SFL still demonstrates a significant performance increase while on Minigrid all methods are very similar (with the robust methods performing slightly better for low values of $\alpha$). Due to the challenging dynamics of JaxNav, even levels with a small number of obstacles can present difficult control and navigation problems meaning ACL methods (such as SFL) still lead to a performance differential over DR. Meanwhile, in Minigrid, due to its deterministic dynamics, difficulty is heavily linked to the obstacle count as this allows for more complex mazes. As such, DR is competitive to ACL methods in settings with fewer obstacles.

## G.3 Analysing the Learnability of Levels

Table 5 shows the mean and median of learnability and success rate for a variety of methods. We find that **the average learnability of levels in the PLR/ACCEL buffers is very low**. While not shown

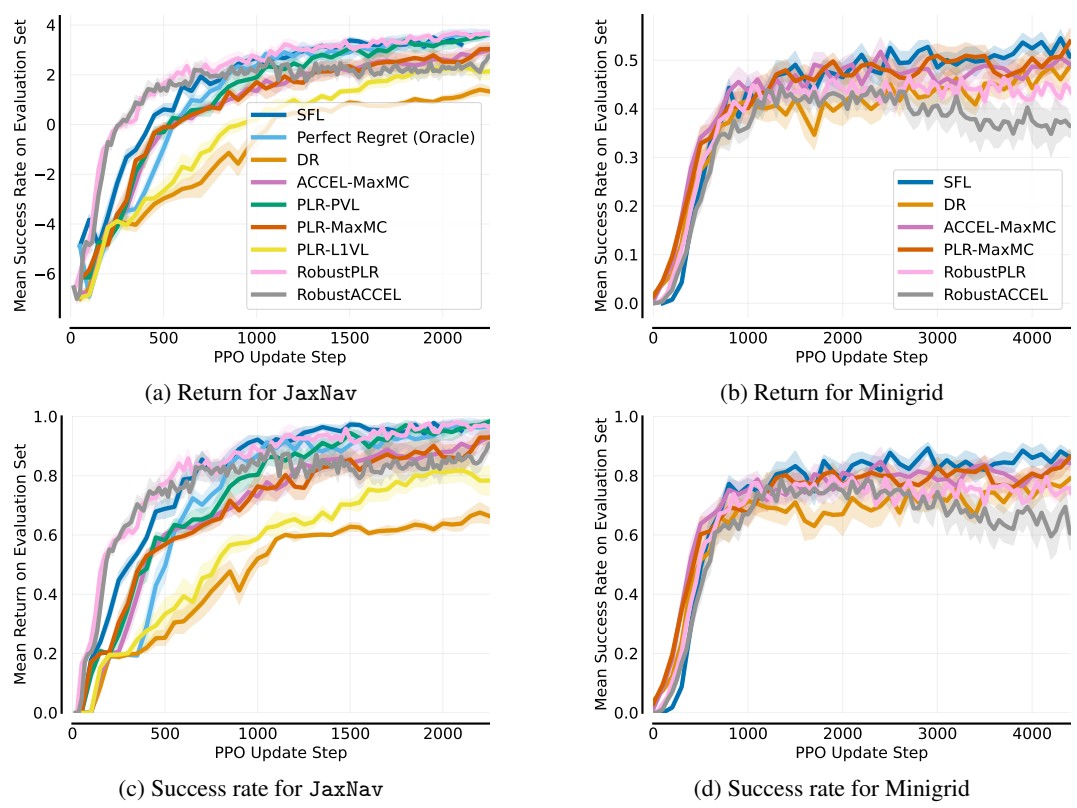

(a) Return for JaxNav

(b) Return for Minigrid

(c) Success rate for JaxNav

(d) Success rate for Minigrid

Figure 16: Episodic return (top) and success rate (bottom) plots for Jaxnav (left) and Minigrid (right).

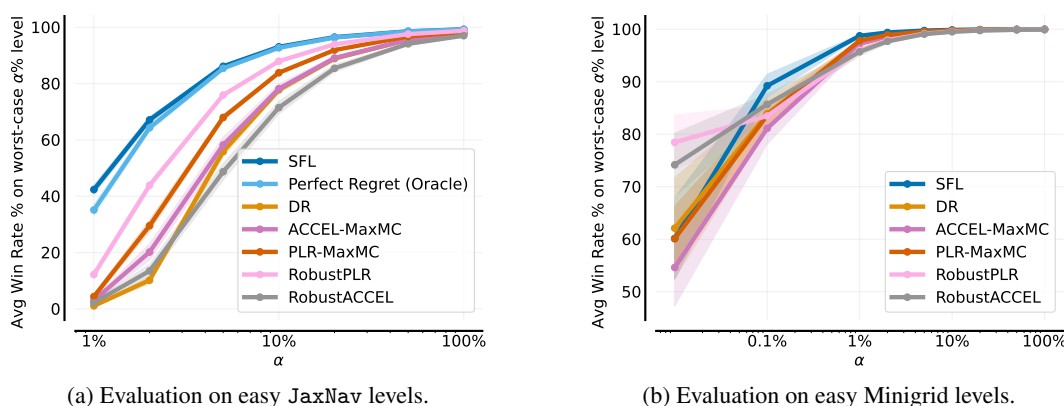

(a) Evaluation on easy JaxNav levels.

(b) Evaluation on easy Minigrid levels.

Figure 17: CVaR evaluation on easy levels for single-agent JaxNav and Minigrid.

here, this is also true when selecting only the levels with the top 50 PVL/MaxMC scores. We also find no statistically significant correlation between the learnability of these levels and the PVL/MaxMC scores. We further note that most of the levels in UED buffers can already be solved by the agent.

By contrast, the levels that SFL selects have a high learnability (note that 0.25 is the maximum learnability value), and a solve rate around 50%. This shows that SFL indeed selects levels with high learnability.

Table 5: The learnability and success rates for levels within the PLR/ACCEL/SFL buffers averaged over training. At each evaluation step, the average and median values for the entire buffer are calculated and then averaged over training. The mean and standard deviation across three different seeds are reported.

| Method | Learnability (Mean) | Learnability (Median) | Success (Mean) | Success (Median) |
|---|---|---|---|---|
| PLR(PVL) | 0.01 (0.00) | 0.00 (0.00) | 0.85 (0.04) | 0.96 (0.05) |
| PLR(MaxMC) | 0.02 (0.00) | 0.00 (0.00) | 0.84 (0.03) | 0.98 (0.01) |
| ACCEL(MaxMC) | 0.01 (0.00) | 0.00 (0.00) | 0.97 (0.01) | 1.00 (0.01) |
| ACCEL(PVL) | 0.01 (0.00) | 0.00 (0.00) | 0.94 (0.04) | 0.95 (0.05) |
| SFL (All Sampled) | 0.01 (0.00) | 0.00 (0.00) | 0.69 (0.01) | 0.99 (0.02) |
| SFL (Selected) | 0.22 (0.01) | 0.22 (0.01) | 0.59 (0.02) | 0.61 (0.03) |

## G.4 Environment Metrics

For single-agent `JaxNav`, we plot the shortest path length, number of walls and the solvability of levels in the PLR/SFL buffers in Figure 18. We find that SFL has marginally longer shortest paths and marginally fewer walls. The SFL levels are also considerably more solvable.

Aside from solvability, there is not a large difference in these metrics between PLR and SFL, despite SFL significantly outperforming PLR. Qualitatively, we find that in `JaxNav`, levels with high learnability tend to involve a lot of turning and intricate obstacle avoidance (as opposed to long paths). As such, the number of walls and shortest path length do not fully capture a level's difficulty.

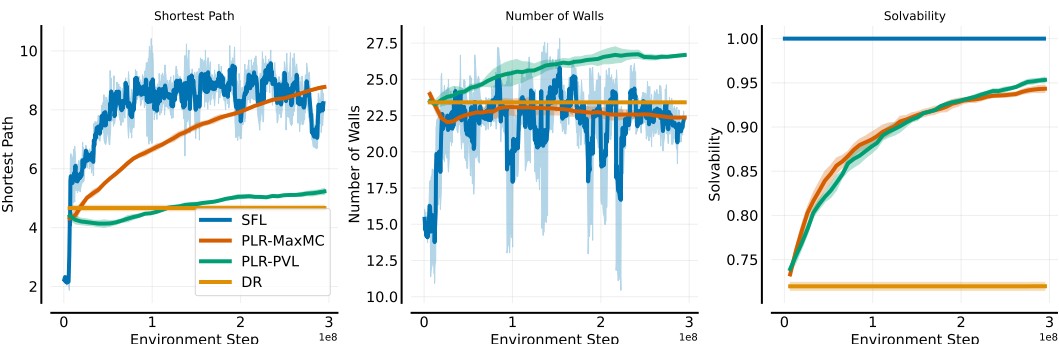

Figure 18: Environment Metrics for single-agent `JaxNav` for SFL, PLR and DR.

For XLand-Minigrid, we report the mean number of rules for the PLR/SFL buffer rulesets in Figure 19, which illustrates a significant difference in the rulesets seen by the different agents. SFL samples well below the mean value throughout training, whereas PLR starts on par with DR before tending easier as training progresses. This result, coupled with the performance difference, illustrates how SFL's learnability score allows it to find the frontier of learning, leading to more robust agents.

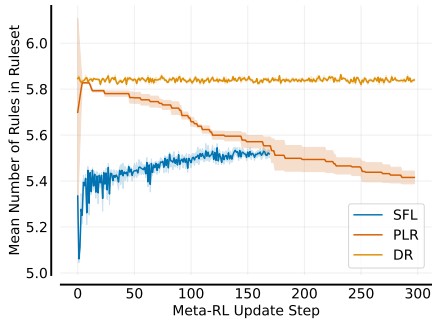

Figure 19: Environment Metrics for XLand-Minigrid.

## G.5 Level Plots

Here we plot some generated levels for single-agent `JaxNav` in Figure 20, multi-agent `JaxNav` in Figure 21 and Minigrid in Figure 22. Overall, ACCEL tends to have the most walls, since its mutation operator is able to add more walls over time. Despite this, the levels do not involve a large amount of complicated turning and maneuvering. The levels selected by SFL, on the other hand, tend to involve going around many corners, which is an important part of `JaxNav`. In multi-agent `JaxNav`, SFL occasionally generates levels where not all of the agents can reach their goals; however, these are still useful to learn on, as the other agents can complete their tasks.

For completeness, we include levels from XLand in Figure 23 but, unlike the other domains, it is difficult to assess a level's difficulty solely from a render.

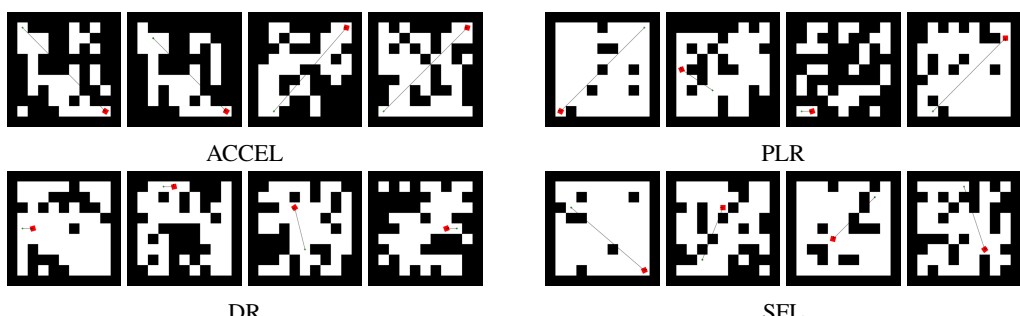

Figure 20: Levels in single-agent `JaxNav` generated by each method.

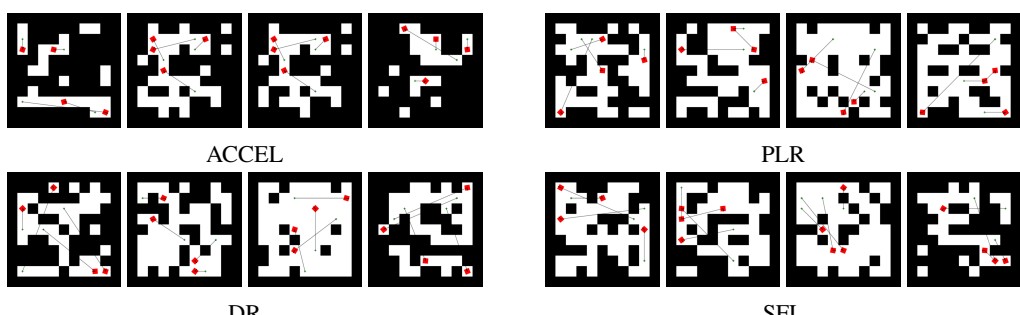

Figure 21: Levels in multi-agent `JaxNav` generated by each method.

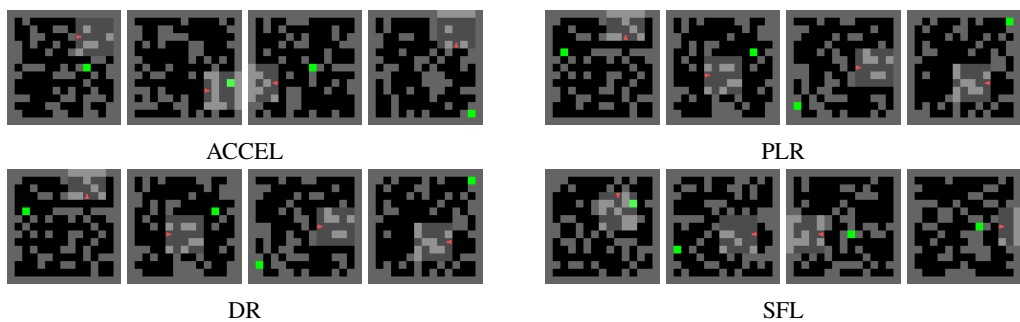

Figure 22: Levels in Minigrid generated by each method.

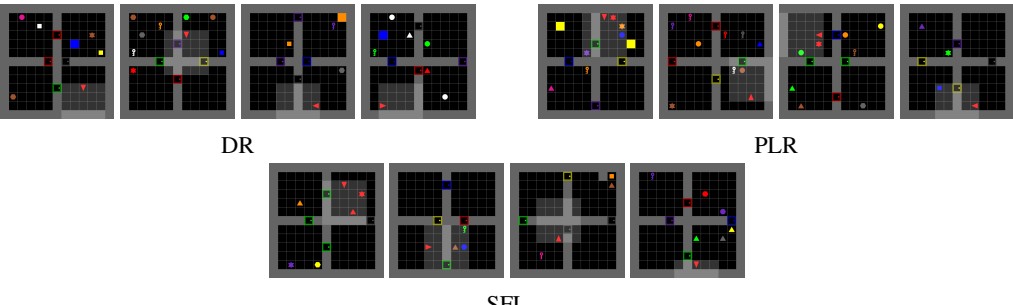

Figure 23: Levels in Xland-Minigrid generated by each method.

# H  Timing Results and Speed Analysis

Tables 6 and 7 report compute time for all methods on single-agent `JaxNav` and Minigrid, respectively. Each individual seed was each run on 1 Nvidia L40s using a server which has 8 NVIDIA L40', two AMD EPYC 9554 processors (128 cores in total) and 768GB of RAM. These times are without logging, and we find that with logging, SFL is around 6% slower than ACCEL on single-agent `JaxNav`. Therefore, for the results in Figure 17b, we use 6% fewer environment steps for SFL to ensure a fair comparison. For Minigrid, SFL is as fast or slightly faster than the other methods. As this is surprising (since SFL performs significantly more environment rollouts), we investigate this further. In Table 8, we compare the time it takes for a single iteration (including training and evaluation) in Minigrid on an L40s GPU.

We note that the SFL rollouts are fast for two reasons:

1. We aggressively parallelise them, running up to 25, 000 environments in parallel, which takes about the same time as running only hundreds in parallel.

2. We do not compute any gradients for these transitions.

The effect of this is that these additional rollouts take significantly less time than the actual training step. Furthermore, UED's training step is more complex than SFL's, since it must maintain a buffer of levels, compute the scores during training, and potentially update the buffer.

The multi-agent results were run on a variety of machines, including the aforementioned L40s system, a similar system featuring NVIDIA A40's and a workstation containing 2 RTX 4090's. On a 4090, a SFL run takes 1d 1h 13m 54s while ACCEL takes 18h 17m 26s.

All XLand-Minigrid experiments were run on 1 Nvidia L40s, with the same server specification as mentioned above. Table 9 reports the compute time for all methods; these times are with logging where each method logs the same data. Note that SFL was compute-time matched to PLR for this environment.

Table 6: Mean and standard deviation of time taken for single-agent `JaxNav` over 3 seeds.

| Method | Compute Time |
|---|---|
| DR | 0:41:33 (0:00:27) |
| RobustACCEL | 1:37:59 (0:00:30) |
| RobustPLR | 1:23:48 (0:00:26) |
| ACCEL | 0:42:09 (0:00:18) |
| PLR | 0:41:48 (0:00:25) |
| SFL | 0:45:45 (0:00:00) |

Table 7: Mean and standard deviation of time taken for Minigrid over 3 seeds.

| Method | Compute Time |
|---|---|
| DR | 0:28:11 (0:00:00) |
| RobustPLR | 0:39:17 (0:00:04) |
| RobustACCEL | 0:33:39 (0:00:04) |
| PLR | 0:29:19 (0:00:00) |
| ACCEL | 0:29:28 (0:00:00) |
| SFL | 0:28:32 (0:00:00) |

Table 8: PLR and SFL timings for a single minigrid iteration

|  | PLR | SFL |
|---|---|---|
| Train Step | 37.5s | 35s |
| Get Learnable Levels | 0 | 2.2s |
| Eval Step | 0.7s | 0.7s |
| **Total** | 38.2s | 37.9s |

Table 9: Mean and standard deviation of time taken for XLand-Minigrid over 5 seeds.

| Method | Compute Time |
|---|---|
| DR | 4:43:06 (0:00:20) |
| PLR | 4:51:47 (0:00:32) |
| SFL | 4:51:16 (0:00:20) |

## I    Ablations

This section of our appendix focuses on running ablations for SFL, investigating the effect of hyperparameters in Appendix I.1, using learnability as a score function in UED in Appendix I.2, and other definitions of learnability in Appendix I.3

### I.1    SFL Hyperparameters

In this section we investigate the effects of SFL's hyperparameters. For computational reasons, we run these ablation experiments over three seeds; furthermore, we consider only `JaxNav`'s single agent setting. Our results, for single-agent `JaxNav` and Minigrid, respectively, are shown in Figures 24 and 25, with the effects of the hyperparameters as follows:

**Number of Sampled Levels** $N$  Sampling more levels results in improved performance, but increases computation time.

**Rollout Length** $L$  For `JaxNav`, there is a small reduction in performance by using a rollout length of 1000, but no gain from using 4000 compared to 2000. In Minigrid, all values perform roughly the same, possibly due to the shorter episodes.

**Buffer Size** $K$  In `JaxNav`, a smaller buffer outperforms larger ones, whereas in Minigrid, there is no significant difference between $K = 100$ and $K = 1000$. This could relate to how easy the environment is (corresponding to how long it takes to learn a particular level). As JaxNav is harder than Minigrid (as it also involves low-level continuous control obstacle avoidance in addition to maze solving), it may be that training on each level more times is beneficial.

**Buffer Update Period** $T$  Increasing the time between updating the set of training levels reduces performance. However, sampling more often increases the computational load of the additional SFL rollouts.

**Sampled Ratio** $\rho$  In `JaxNav`, a higher sampling ratio seems preferable, similarly to the "robust" version of PLR [4]. In Minigrid, however, using $\rho = 0.5$ performs slightly better than $\rho = 1$.

**Sampling in decreasing order of learnability**  For `JaxNav`, we trialled selecting levels in decreasing order of learnability rather than randomly among the top $K$. This performs better, and has roughly the same effect as reducing the buffer size. This makes sense, as by reducing the buffer size or selecting in decreasing order, we are restricting the range of levels that can be chosen to only the highest-learnability ones.

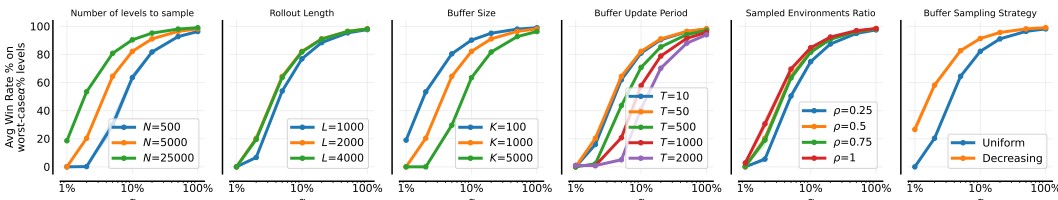

Figure 24: Analysing the effect of hyperparameters on single-agent `JaxNav`. Hyperparameters not mentioned in each plot use the default configuration's values: $N = 5000$, $T = 50$, $\rho = 0.5$, $K = 1000$, $L = 2000$.

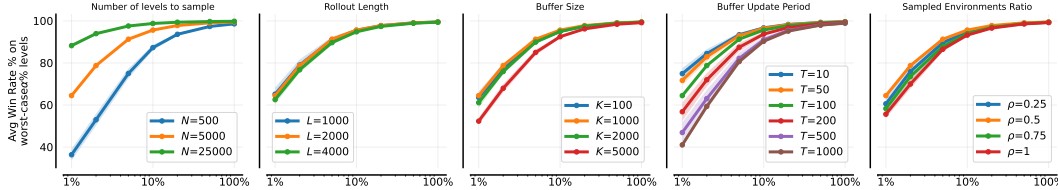

Figure 25: Analysing the effect of hyperparameters on Minigrid. Hyperparameters not mentioned in each plot use the default configuration's values: $N = 5000$, $T = 100$, $\rho = 0.5$, $K = 1000$, $L = 2000$.

## I.2 UED with learnability as a score function

We now apply our learnability metric as a score function for UED. In particular, when performing our DR rollouts, we compute the learnability as in Section 4.3 and use that as the score function. Figure 26 shows these results; Learnability improves performance compared to MaxMC, supporting our claim that the score function is the primary limitation of current UED methods.

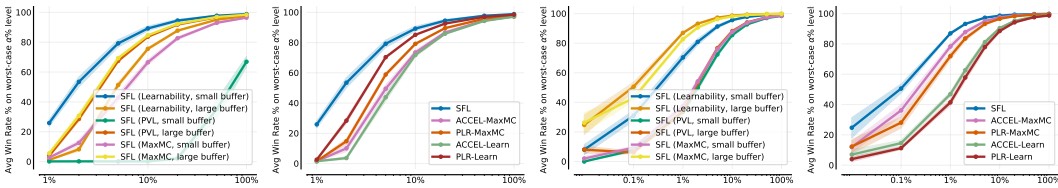

(a) PVL as SFL's score function in `JaxNav`.

(b) Learnability as UED's score function in `JaxNav`.

(c) PVL as SFL's score function in Minigrid.

(d) Learnability as UED's score function in Minigrid.

Figure 26: Comparing SFL as a score function for UED and vice-versa (for single-agent `JaxNav` and Minigrid). In Figures (a, c), the large buffer is of size 1000 while the small is of size 100. We find that **SFL with learnability outperforms all other combinations.**

## I.3 Different Definitions of Learnability

In this section, we investigate alternative definitions of learnability. All of the following functions have a learnability value of $0$ for a success rate of $0.0$ and $1.0$, but differ in how scores are assigned to intermediate success rates $p$. We aim to assess whether our learnability score's main contribution is, in fact, just excluding unsolvable and perfected levels; our analysis indicates that this is not the case.

First, on `JaxNav`, we investigate the case where the peak learnability is not at $p = 0.5$, but at other values. To do this, we represent the learnability function as a piecewise quadratic, see Figure 27a for an illustration. During this test, the peak learnability value remained at $0.25$ but we varied the success rate at which the peak occurs. The results are in Figure 27b, and we find that a peak of $0.6$ performs the best, and that performance slightly degrades as the peak moves towards higher success rates. In particular, performance is poor when the peak is at $p = 0.99$.

In Figures 27c and 27d, we consider other definitions of learnability on `JaxNav` and XLand. We keep the restriction that learnability equals zero when $p = 0$ or $p = 1$. The first is *Uniform*, where all levels with success rates $0 < p < 1$ are assigned an equal score. *Linear*(0) has learnability linearly increasing from $1.0$ to $0.0$, and *Linear*(1) has learnability linearly increasing from $0.0$ to $1.0$.

We find that *Linear*(0) (which is an approximation of true regret) performs similarly to the default definition of learnability on `JaxNav` but struggles on XLand. Meanwhile, Uniform sampling performs worse than our approach but still outperforms all UED methods on `JaxNav` and XLand (albeit by a smaller margin than SFL). Finally, *Linear*(1) performs significantly worse on `JaxNav` but is the closest to SFL's performance on XLand. These results indicate that our chosen definition of the learnability score function is more robust than these alternatives across different domains.

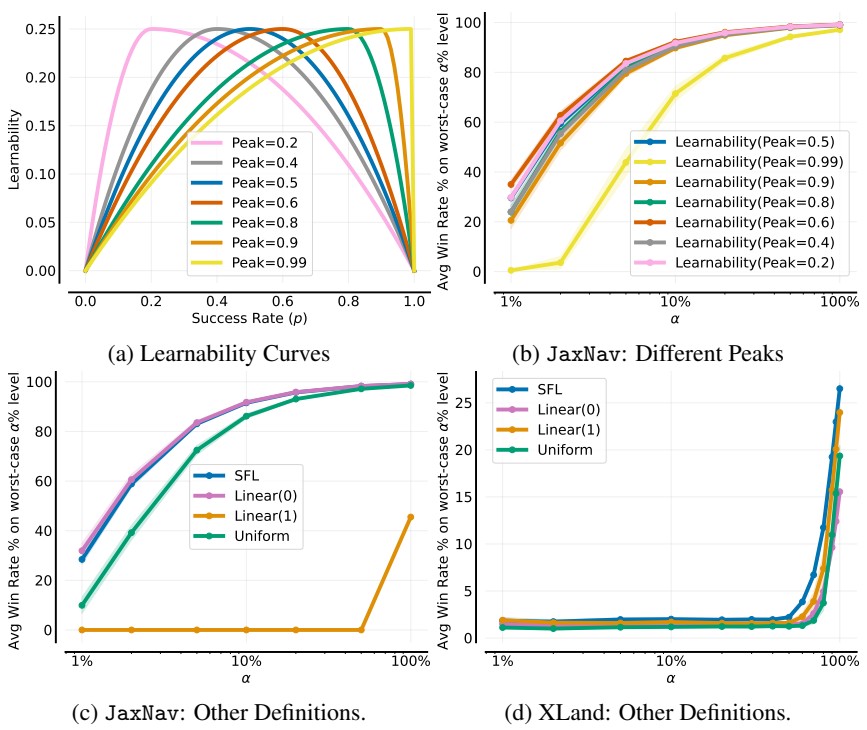

(a) Learnability Curves

(b) `JaxNav`: Different Peaks

(c) `JaxNav`: Other Definitions.

(d) XLand: Other Definitions.

Figure 27: Other learnability definitions.

