# OpenReview forum: "No Regrets: Investigating and Improving Regret Approximations for Curriculum Discovery"
_NeurIPS.cc/2024/Conference — NeurIPS 2024 poster_

### Official Review · Reviewer_ycoo · 2024-07-03

**Soundness:** 2
**Presentation:** 1
**Contribution:** 3
**Rating:** 5
**Confidence:** 4

**Summary:**

The paper investigates the limitations of popular UED approaches, such as PLR and ACCEL, demonstrating that they do not improve upon the Domain Randomisation baselines, where levels are randomly sampled. The author's main claim is that learnability does not correlate with the scoring functions MaxMC and PVL used by current UED approaches, leading to sub-optimal performance.

To address this issue, they propose Sampling For Learnability (SFL), a simple algorithm where levels are chosen to prioritize learnability. In the context of the paper, the learnability of a level is defined as $p \cdot (1-p)$, where p is the success rate of the agent on that given level.
The authors conduct experiments on three different environments: MiniGrid, Single, and Multi-Agent JaxNav. As a metric, they propose the use of the conditional value at risk (CVaR), evaluating on an α% of levels on which the agent performs worst, from a randomly generated buffer of levels.

The results show that SFL outperforms current UED approaches. Simultaneously, when it is possible to compute it, true regret outperforms SFL, indicating that the flaw of UED algorithms is indeed in the scoring function approximation. To summarize, the main contributions of the authors are:

- Demonstrate sub-optimality of current UED approaches, due to lack of correlation between their scoring function and learnability
- Propose a new algorithm, SFL, which prioritizes levels with high learnability

**Strengths:**

- UED has indeed been seen to perform sub-optimally in practice, i.e. as evidenced in the JaxUED paper. It is thus an important problem to address since limited work has been done

- The investigation of the correlation between learnability and the scoring function is a good justification for the paper itself to exists, and gives a plausible reason for the poor performances of UED on some tasks.

- The SFL algorithm proposed is novel (as far as I know), simple, and easy to understand. The results obtained on the 3 environments they considered seem promising.

- CVaR seems a more principled and justified way to investigate the robustness of Auto-curricula algorithms

**Weaknesses:**

- Learnability definition: why are environments that are solved exactly half of the time the most valuable ( given the $p(1-p)$ function used to define it)? It seems an arbitrary choice, which should need to be justified better through additional experiments. I do not agree that levels that are solved 5% of the time are as valuable as ones solved 95% for example.

- Many parameter choices are just reported with no justification supporting it ( e.g. hyperparameter $\rho$ for the algorithm)

- Experiments are poor in broadness. Given the purely empirical nature of the paper, I would expect way more experiments to indeed show that SFL outperforms the UED baselines. This includes more seed for the Multi-agent JaxNav, and more environments. As of now, it is not possible to actually confirm that SFL is indeed better than UED algorithms.

- The writing is poor and sometimes confusing. Learnability is referenced way before being formally defined, the syntax of some sentences is poor, etc... By reading the paper, it seems the writing has been rushed.

**Questions:**

- In Algorithm 1, $\rho$ is set to $0.5$. Hence, half of the levels used for updating the policy $\pi_\phi$ are ones with high learnability, and half are sampled at random. Why do not set $\rho$ to a higher value? I would like to see a plot showing the performance of SFL upon varying $\rho$, so as to have a better idea of how indeed training in 'more learnable' environments correlates with better performances

- In Alg.1, what are the performances if instead of sampling uniformly from the high-learnable levels $\mathcal{D}$, you select levels in decreasing order of learnability?

- What is the average/median learnability of the randomly created levels in Alg 2? What about the top-K ones? It is important to see and report it so indeed one is sure that the levels you are using are indeed ones with high-learnability. As far as I can infer from the paper, it could still be that most of the randomly created levels have low learnability.

- Can you provide more details on what you did to make current UED methods work better? The poor performance of UED algorithms was already highlighted in the JAXUED paper, but I am not sure if this is just because they were not optimized fully, e.g. via a more extensive hyper-parameter search

- Did you try using different definitions of Learnability? How did they perform?

**Limitations:**

No ethical limitations

---

> ### Author Rebuttal · Authors · 2024-08-06
>
> We appreciate your thorough review and useful comments, particularly highlighting that we are focusing on an "important problem", and that our algorithm is "simple, and easy to understand". Please find our responses below.
>
> # Weaknesses
> ## Different Definitions of Learnability
>
> Intuitively, we justify our definition as follows (in a goal-based setting where there is only a nonzero reward for reaching the goal):
> - $p$ represents how likely the agent is to obtain positive learning experiences from a level.
> - $1-p$ represents the maximum potential improvement the agent can make on that level.
> - Multiplying these yields (probability of improvement) * (improvement potential), i.e., expected improvement.
>
>
> $p(1-p)$ can also be seen as the variance of a Bernoulli distribution with parameter $p$, i.e., how inconsistent the agent's performance is.
>
> We have added this explanation to the updated manuscript to increase clarity.
>
> To confirm this choice we have run a set of experiments on single-agent JaxNav where we represent the learnability function as a piecewise quadratic. During this test, the peak learnability value remained at 0.25 but we varied the success rate at which the peak occurs. The results are in Figure 4\(c) of the attached PDF; overall, a peak around $p=0.5$ performs the best, and moving away from this reduces performance slightly. However, as performance is not dramatically worse, this indicates that the most important aspect of learnability is in rejecting levels with perfect or zero success rates.
>
> ## Hyperparameter choices / Ablations
>
> Thank you for raising this point. We have tuned all of the hyperparameters, and have run ablations on SFL. **Our conclusions remain unchanged (i.e., that SFL outperforms UED)**, but we have more insights now into which parameters are most important for SFL. Furthermore, we find that SFL is less sensitive to hyperparameters than PLR or ACCEL. The new tuned results are in Figure 1, and the ablations are in Figure 2 of the 1-page PDF.
>
> ## More seeds and environments
>
> Thank you for raising this point. We have run more seeds for multi-agent JaxNav, bringing the total count to 5, and our results have not changed. We have also included results on an additional environment, XLand-MiniGrid, as described in our global response. Here, SFL also outperforms PLR and DR.
>
> We now **have results in four environments demonstrating SFL's superiority compared to SoTA UED methods**. We further compare against several ablations. We believe these results are sufficiently broad to showcase SFL's performance.
>
>
> ## Writing
>
> Thank you for raising the writing quality issues, and we apologise for letting them slip through. We have carefully reviewed the submitted manuscript and have fixed these issues for our updated version.
>
> # Questions
> ## Different values of the sampling ratio $\rho$
>
> We agree that this is an important question and we have tested various values of $\rho$ on single-agent JaxNav with results shown in Figure 2 of the attached PDF. Increasing $\rho$ beyond 0.5 increases performance slightly.
> Due to the high maximum fill % used for JaxNav, we hypothesise that the randomly generated levels are able to present challenging control and navigation problems.
> While increasing $\rho$ makes the agent train on more learnable levels, it may also decrease the diversity of the levels seen by the agent.
>
> ## Decreasing order of learnability
>
> Thank you for this suggestion, we agree it is a useful question to answer. We have trialled selecting levels in decreasing order of learnability rather than randomly and, as illustrated in Figure 1 of the attached PDF, this has roughly the same effect as reducing the buffer size.
> This makes sense, as by reducing the buffer size or selecting in decreasing order, we are restricting the range of levels that can be chosen to only the highest-learnability ones.
>
> ## Average vs top-k learnability
>
> For 45 iterations for single-agent JaxNav, we logged the average and median learnability of the entire set of sampled levels, and the subset we use to train on, with results below. In short, **the entire set of sampled levels generally has low learnability, whereas the top 100 levels have much higher values**. This shows that SFL is filtering out levels with very low learnability (i.e., levels that the agent either already perfectly solves or that are currently too hard). This result further explains why we outperform existing methods. We appreciate you raising this point and have added this analysis to the manuscript.
>
> |                  | Mean All   | Median All | Mean Subset | Median Subset |
> |------------------|------------|------------|-------------|---------------|
> | Value            | 0.014 | 0.0        | 0.198  | 0.195    |
>
>
> ## What we did to make UED better
>
> For the existing UED methods, we used the open-source implementations provided by JaxUED and conducted a thorough sweep of hyperparameters, as explained in our general response. Additionally, as JaxUED showed, and as we've also found, DR can be surprisingly competitive if the PPO parameters are tuned appropriately. Indeed, in the multi-agent setting of JaxNav, DR is in line with or outperforms all existing UED methods during CVaR evaluation; albeit the UED methods use hyperparameters tuned on the single-agent JaxNav setting.
>
> ## Different learnability definitions?
>
> Following your suggestion, we have experimented with different peak values, as outlined above. We have not tried an altogether different definition because ours followed from the intuition explained above. However, our code will be open-sourced, allowing the community to easily try alternative definitions.
>
>
> # Conclusion
> Thank you once again for your detailed review. We hope our responses have addressed your concerns, and we welcome any further discussion or questions you might have. If we have successfully alleviated your concerns, we kindly ask you to consider updating your score to recommend accepting our paper.

---

> > ### Comment · Reviewer_ycoo · 2024-08-11
> > **Thanks for the answer**
> >
> > Thanks for your detailed answer. I took some more time to think about the points you raised:
> >
> > **Different Definitions of Learnability**: I see that indeed rejecting impossible or already completely solved level is indeed the biggest factor in increasing performance. I would still argue however that I find the learnability definition not very principled - if what mentioned in the first sentence is true, many other scoring function which do the same in those extreme cases, and which chosoe a better heuristic in the remaining scenarios will outperform SFL. Isn't that the natural thing to try?
> >
> > **Hyperparameter choices / Ablations**: Can you clarify the sentence ' we have more insights now into which parameters are most important for SFL'. Which are these hyperameters, and more specifically, why is that the case?
> >
> > **More seeds and environments**: Thanks for adding more seeds and an additional environment.
> >
> > **Writing**: can you provide the most significant changes done to the writing?
> >
> > **Different values of the sampling ratio**:   You say *increasing makes the agent train on more learnable levels, it may also decrease the diversity of the levels seen by the agent* - do you have any insight on the structure of the levels that are create by SFL, especially when $\rho$ is increased to 1? Many similar plots have been produced by other similar papers, such as Fig. 3 in the ACCEL paper (https://arxiv.org/pdf/2203.01302).
> >
> > **Average vs top-k learnability**: You should also compare this with the levels of learnability in the buffers of PLR and ACCEL for example, such as one can have a complete picture of where the advantage by using SFL is coming from.
> >
> > **What we did to make UED better** and **Different learnability definitions?**: Thanks for your answer
> >
> > In general, I agree with the authors when saying that SFL outperforms UED method in the environments considered. However, beside the remaining doubts above, I am still on the fence for this work. My main criticism comes from the fact that a new method (SFL) is proposed, but that there is no extensive justification on why this is happening. I believe it would be good to have more insight in the structure of the levels SFL prioritizes, and if there are other gaps SFL is actually filling beside excluding impossible or completely solved levels. Is learnability itself better, or any method which excludes the pathological type of levels I just mentioned would actually perform similarly? I believe this is an important point to address.
> >
> > Sorry for the delay in the answer, and I understand if you are not able to produce a response in time!

---

> > > ### Author Response · Authors · 2024-08-12
> > > **Response 1/2**
> > >
> > > Dear reviewer, thank you for your response, your continued engagement, and your commitment to improving our paper! Please find our responses below.
> > >
> > >
> > > # Hyperparameter choices / Ablations
> > > First, as illustrated in Figure 2 in the rebuttal PDF, SFL is robust to all of its hyperparameters, and suboptimal choices do not lead to catastrophic failure.
> > >
> > > That being said, the most important hyperparameters are:
> > > - $N$: How many levels to sample. Sampling more levels increases performance, as more levels gives SFL a larger sample to draw high-learnability levels from.
> > > - $K$: Buffer Size. In Jaxnav, we find a small buffer size is beneficial, and in Minigrid it is better to have a slightly larger one. This could relate to how easy the environment is (corresponding to how long it takes to learn a particular level). As JaxNav is harder than Minigrid (as it also involves low-level continuous control obstacle avoidance in addition to maze solving), it may be that training on each level more times is beneficial. By contrast, since Minigrid is easier, the number of episodes required in a level may be much less.
> > > - $T$: The buffer update period. If this value is too large, then performance degrades. This is because the learnability of the buffer, and therefore the usefulness of the levels, decreases as the agent trains on it.
> > >
> > >
> > > # Writing
> > >
> > > We kept the structure of our manuscript largely the same and made several local and low-level writing changes and fixes to ensure the writing flows better. One structural change we did make is to define learnability earlier on, in Section 4.1. We finally also added additional explanations and descriptions based on the reviews by yourself and the other reviewers.
> > >
> > > # Different values of the sampling ratio
> > > We agree such a plot would be useful, and will provide one in our revised manuscript.
> > > Since we are unable to share figures during the discussion period, we report the shortest path, number of walls, and solvability, averaged over training in the table below.
> > > We find that SFL has marginally longer shortest paths and marginally fewer walls. The SFL levels are also considerably more solvable.
> > >
> > > | Method     | Shortest Path (Mean)   | N Walls (Mean)   | Solvable (Mean)   |
> > > |:-----------|:-----------------------|:-----------------|:------------------|
> > > | PLR (PVL)   | 4.74 (0.06)            | 25.59 (0.19)     | 0.89 (0.01)       |
> > > | PLR (MaxMC) | 7.03 (0.49)            | 22.63 (0.21)     | 0.89 (0.02)       |
> > > | SFL        | 8.12 (0.02)            | 21.98 (0.43)     | 1.00  (0.00)       |
> > >
> > >
> > > Aside from solvability, there is not a noticable difference in these metrics between PLR and SFL, despite SFL significantly outperforming PLR. Qualitatively, we find that in JaxNav, levels with high learnability tend to involve a lot of turning and intricate obstacle avoidance (as opposed to long paths). As such, the number of walls and shortest path length do not fully capture a level's difficulty. We will add illustrative examples of generated levels to our Appendix to demonstrate this point.
> > >
> > > # Average vs top-k learnability in UED:
> > > Thank you for suggesting this experiment, we have run the same analysis for ACCEL and PLR in JaxNav. We further compute the correlation between the score (e.g. MaxMC or PVL) and learnability for all levels in the buffer. We find that:
> > > - **The average learnability of levels in the PLR/ACCEL buffer is very low.** This is also true when selecting only the levels with the top 50 PVL/MaxMC scores.
> > > - There is no significant correlation between PVL/MaxMC and learnability.
> > > - Most of the levels selected by PVL and MaxMC can consistently be solved by the agent.
> > >
> > > The table below reports the learnability and success rates for levels within the PLR/ACCEL buffers averaged over training. At each evaluation step, we calculate the average and median values for the entire buffer and then average these values over training. Finally, we report the mean and standard deviation across three different seeds.
> > >
> > > | Method       | Learnability (Mean)   | Learnability (Median)   | Success Rate (Mean)   | Success Rate (Median)   |
> > > |:-------------|:----------------------|:------------------------|:-----------------|:-------------------|
> > > | PLR (PVL)     | 0.01 (0.00)           | 0.00 (0.00)             | 0.85 (0.01)      | 0.96 (0.00)        |
> > > | PLR (MaxMC)   | 0.02 (0.00)           | 0.00 (0.00)             | 0.85 (0.03)      | 0.95 (0.04)        |
> > > | ACCEL (MaxMC) | 0.02 (0.00)           | 0.00 (0.00)             | 0.93 (0.01)      | 0.97 (0.02)        |
> > > | ACCEL (PVL)   | 0.01 (0.00)           | 0.00 (0.00)             | 0.94 (0.02)      | 0.97 (0.02)        |
> > >
> > >
> > > This result further strengthens our findings (shown in Figure 2 of the original manuscript) that current UED score functions do not correspond to learnability. Instead, most levels in the UED buffers can already be solved 100% of the time. By contrast, as shown in our rebuttal, SFL consistently selects levels with high learnability.

---

> > > > ### Author Response · Authors · 2024-08-12
> > > > **Response 2/2**
> > > >
> > > > # Justification for SFL & Different Definitions of Learnability
> > > > We believe the intuition for learnability is simple, easy to understand, and our score function works well empirically compared to prior approaches. However, we are not claiming that learnability is the be-all and end-all for score functions; we fully agree that there may be other variations of it that could perform better, and we are glad the reviewer raised this important point for future work.
> > > >
> > > > Furthermore, we would like to point out that the regret approximations used in current UED methods (e.g., MaxMC and PVL) are not particularly principled and are at best a loose approximation to regret. Indeed, in [1], six possible functions are trialled with their final choice based solely on empirical results. Prior work has also not investigated how closely these metrics correlate to true regret, and recent work has not investigated any alternatives to PVL and MaxMC.
> > > >
> > > > Further justification for our score function is presented below in three parts:
> > > >
> > > > **First, we wish to emphasise that our learnability definition is deeply connected to a large body of prior work in learning theory and curriculum development.** Intuitively, many curriculum methods (including SFL) aim to obtain experiences within the agent's "zone of proximal development", i.e., those that are neither too easy nor too hard [2].
> > > >
> > > > Works such as [3] use hardcoded reward thresholds to define a goal that is of appropriate difficulty. Similarly, POET [4] also uses predefined reward thresholds to filter environments. SFL, meanwhile, directly uses the success rate of the agent on the level to compute the score.
> > > >
> > > >
> > > > **Second, we believe our empirical results are a strong justification of SFL**, with the reviewer agreeing that "SFL outperforms UED". We would also note that most prior peer-reviewed work in UED uses two ([5, 6]) or three ([7]) environments, and we have strong empirical results in four.
> > > >
> > > >
> > > > **Third, we have run several other variations on learnability, all of these assign a level a score of zero when $p=0$ or $p=1$** (in JaxNav):
> > > > 1. Uniform sampling of levels with $0 < p < 1$.
> > > > 2. Learnability linearly increasing from $p=1$ to $p=0$.
> > > > 3. Learnability linearly increasing from $p=0$ to $p=1$.
> > > >
> > > > We find that variation 2. (which is an approximation of true regret) performs similarly to the default definition of learnability.
> > > > Variation 1. performs worse than our approach, but still outperforms all UED methods (albeit by a smaller margin than SFL).
> > > > Finally, variation 3. performs significantly worse, and we believe this is due to this score function prioritising easy levels, which provide less opportunity for the agent to improve. We will include these results in our updated Appendix.
> > > >
> > > >
> > > > While we focused on $p(1-p)$, these results (and the previous results with different learnability peaks) show that our method performs robustly under other reasonable learnability estimates, demonstrating the potential of learnability in the UED setting and the robustness of SFL.
> > > > While we agree that the exact choice of learnability metric is still an open question, we believe that all our claims are justified empirically, and our results make an impactful contribution to the UED community.
> > > >
> > > >
> > > >
> > > >
> > > > **References**
> > > >
> > > > [1] Jiang, Minqi, et al. "Prioritized level replay." International Conference on Machine Learning. PMLR, 2021.
> > > >
> > > > [2] L. Vygotsky. Interaction between learning and development. Readings on the Development of Children, pages 34–40, 1978.
> > > >
> > > > [3] Carlos Florensa, David Held, Xinyang Geng, Pieter Abbeel. ICML, 2018
> > > >
> > > > [4] Wang, Rui, et al. "Paired open-ended trailblazer (poet): Endlessly generating increasingly complex and diverse learning environments and their solutions." arXiv preprint arXiv:1901.01753 (2019).
> > > >
> > > > [5] Dennis, Michael, et al. "Emergent complexity and zero-shot transfer via unsupervised environment design." Advances in neural information processing systems 33 (2020): 13049-13061.
> > > >
> > > > [6] Jiang, Minqi, et al. "Replay-guided adversarial environment design." Advances in Neural Information Processing Systems 34 (2021): 1884-1897.
> > > >
> > > > [7] Parker-Holder, Jack, et al. "Evolving curricula with regret-based environment design." International Conference on Machine Learning. PMLR, 2022.
> > > >
> > > >
> > > >
> > > > # Conclusion
> > > > Once again, we appreciate the reviewer's engagement in the review process, and their helpful suggestions. We believe these results further strengthen our paper.

---

> ### Comment · Reviewer_ycoo · 2024-08-12
> **Response to authors**
>
> Dear Authors,
> Thanks for your detailed response and additional experiments.
>
> **Different values of the sampling ratio**: Thanks for producing these results. In the final manuscript, I would be interested in seeing other possible features of the environment which are common in levels sampled with SFL
>
> **Average vs top-k learnability in UED**: This result is interesting. I would consider to appropriately insert it in the appendix of the  final version of the paper.
>
> **Justification for SFL & Different Definitions of Learnability**: Thanks for providing additional examples of learnability. This addresses the issues I raised.
>
> Given the above discussion, I believe most of the issues I raised were appropriately addressed by the authors. I will raise my score accordingly.
>
> Thanks again for the detailed answers and the additional experiments, and good luck with the paper

---

> > ### Author Response · Authors · 2024-08-12
> > **Response to ycoo**
> >
> > Dear Reviewer,
> >
> > Thank you for your timely response to our comments. We believe we have addressed the concerns you have raised in detail, including through additional experiments which confirm and strengthen our original findings. We also appreciate the increase in your score but were hoping for more significant support given your positive response.

---

### Official Review · Reviewer_Ns9r · 2024-07-09

**Soundness:** 3
**Presentation:** 3
**Contribution:** 3
**Rating:** 7
**Confidence:** 3

**Summary:**

This experimental paper proposes a UED method (SFL) for JaxNav, a continuous single- and multi-robot navigation task in a partially observable setting.  The authors document the shortcomings of UED methods, (Domain Randomisation, Prioritised Level Replay, and ACCEL) on this partially observed, continuous action, continuous state setting.  The authors provide a demonstration that the scoring mechanism used by the above UED methods is misguided.  Their method, SFL, uses a “learnability score” which focuses learning on levels for which agents achieve success rates closer to 50%.   To better focus on robustness and generalization, the authors develop an empirical CvaR measure of success for evaluation and comparison of methods.

**Strengths:**

Scientists who are interested in developing and implementing the RL agents in real life will find this experimental paper important.

The paper is technically sound and provides simulation support for the claims.   The authors provide a nice discussion of the weaknesses of SFL (proposed version works only for binary outcome, deterministic environment).  I have not gone over the github site;  aside from the points made below, replicability is good.

The learnability score is original (obvious after reading the paper, but perhaps not before).  This score nicely takes advantage of the JaxNav environment which provides fast training of an RL agent.  Related work is adequately cited and it is very clear how SFL differs from the 3 UED methods discussed.

**Weaknesses:**

See questions below.

**Questions:**

Lines 15-17.  Suggest to delete spurious statements such as “We had tried our best to make current UED methods work for our setting before going back to the basics and developing our new scoring function, which took a few days and worked out-of-the-box.  We hope our lessons and learnings will help others spend their time more wisely.”   Don’t complain!

The authors should make an effort to assist readers familiar with RL but not ACL methods by providing details in the appendix.  For example in line 175, Algorithm 1 does not provide information on how \phi is updated.  The update algorithm could be provided in the appendix.

Figure 2 includes statements about “learnability”  prior to a definition of learnability.

Lines 154-157.  Learnability has not yet been defined so statements about slight/no correlation with learnability are vacuous.   It is not clear to reader whether learnability in these lines/Figure 2 is the same as the definition of learnability given later (and used by SLR).

Lines 133-135 might better go in the later section on weaknesses/limitations.

Line 214.  The authors do not define “solvable.”  The reader needs to know how you are operationalizing this term.

Line 274-5.  Unclear what “perfect regret” means.

**Limitations:**

Limitations are nicely discussed.  It would be cool if the authors could comment on how SFL might be generalized to continuous outcomes and stochastic environments.

---

> ### Author Rebuttal · Authors · 2024-08-06
>
> Thank you for your positive review, particularly highlighting that our paper is "technically sound" and that the learnability score is "original". Please find our responses to your comments below.
>
>
> > Lines 15-17. Suggest to delete spurious statements such as “We had tried our best to make current UED methods work for our setting before going back to the basics and developing our new scoring function, which took a few days and worked out-of-the-box. We hope our lessons and learnings will help others spend their time more wisely.” Don’t complain!
>
> We agree and have removed this from our manuscript.
>
> > The authors should make an effort to assist readers familiar with RL but not ACL methods by providing details in the appendix. For example in line 175, Algorithm 1 does not provide information on how \phi is updated. The update algorithm could be provided in the appendix.
>
> Thank you for your suggestion and we agree that we should provide a more detailed background on ACL in our appendix. On line 175, $\phi$ is updated using any RL policy learning algorithm (we use PPO) and as such is not affected by the ACL process, we will make this distinction clearer in our updated manuscript.
>
> > Figure 2 includes statements about “learnability” prior to a definition of learnability
>
> > Lines 154-157. Learnability has not yet been defined so statements about slight/no correlation with learnability are vacuous. It is not clear to reader whether learnability in these lines/Figure 2 is the same as the definition of learnability given later (and used by SLR).
>
> Thank you for pointing this out! We apologise for this oversight and we have rectified this by defining learnability earlier on in our paper.
>
> > Lines 133-135 might better go in the later section on weaknesses/limitations.
>
> We appreciate this suggestion and have implemented this in the updated manuscript.
>
> > Line 214. The authors do not define “solvable.” The reader needs to know how you are operationalizing this term.
>
> We added an explanation to the manuscript., thank you for bringing it to our attention. "Solvable" in our paper means that the goal state can be reached in a particular level, i.e., it is not impossible to complete.
>
> > Line 274-5. Unclear what “perfect regret” means.
>
> Regret is defined as the difference in return between the optimal policy on a level, and the current policy. However, since it is intractable in practice to compute this, most methods use approximations (such as PVL and MaxMC). We use "perfect regret" to refer to the exact implementation of regret, which is sometimes possible (e.g. in gridworlds we can compute the optimal return). We have made this clear in our updated manuscript.
>
> >  It would be cool if the authors could comment on how SFL might be generalized to continuous outcomes and stochastic environments.
>
> Here are just some possible options to explore. We have added this discussion to our updated manuscript, thank you for raising this point as an important step of future work!
>
> - In a non-binary-outcome domain, we could potentially reuse the intuition that $p(1-p)$ can be seen as the variance of a Bernoulli distribution; therefore, in a continuous domain, an analogous metric would be the variance of rewards obtained by playing the same level multiple times.
> - For stochastic environments, we could form an extended level space, where the random seed and the level $\theta$ are bundled into $\tilde \theta$. In this way, we could turn a stochastic environment into a deterministic one, and since we assume simulator access already, this is not a significantly stronger assumption.
>
> # Conclusion
> Thank you again for your helpful review, we hope we have addressed your comments satisfactorily, and welcome further discussion.

---

> > ### Comment · Reviewer_Ns9r · 2024-08-07
> > **Thanks**
> >
> > Thanks for your response.  I will hold to the rating of 7.

---

### Official Review · Reviewer_gAKk · 2024-07-12

**Soundness:** 3
**Presentation:** 3
**Contribution:** 3
**Rating:** 7
**Confidence:** 4

**Summary:**

This work proposes a new Unsupervised Environment Design (UED) method, called Sampling for Learnability (SFL), developed for navigation tasks. SFL upsamples environment configurations with high learnability scores, i.e., p*(1-p), where p is the success rate in a specific configuration. The paper highlights the failure of some of the state-of-the-art (SOTA) UED approaches on such tasks and argues that they cannot approximate learning agents' regret, unlike SFL. They evaluate SFL in JaxNav, a Jax-based single and multi-agent navigation benchmark they introduce, and MiniGrid. Their empirical results evidence that SFL is robust and can outperform Domain Randomisation (DS) and SoTA UED approaches in JaxNav and MiniGrid. The paper demonstrates these outcomes by following two evaluation protocols, respectively: 1) A risk-based protocol that they introduce, computing the conditional value-at-risk (CVaR) of the distribution of success rate based on randomly sampled levels, and 2) a common protocol in the UED literature that evaluates performance/success in hand-designed test sets, as they focus on complex, yet, solvable configurations.

**Strengths:**

- This paper is well-written, clearly expresses the motivation and the gap in the literature, and illustratively analyses the failure of existing UED methods in navigation tasks.

- Their contribution is a simple yet novel UED approach called SFL that explicitly addresses this failure mode. Their empirical results through two evaluation protocols highlight that SFL is robust (in terms of CVaR-based metrics) and outperforms existing approaches in expected return/success.

- I agree with the authors' claim that existing UED methods claim robustness yet fail to quantify it accurately. Hence, the robustness evaluation protocol utilized in this work may be considered new in the UED literature.

**Weaknesses:**

- Section 4.1 analyses MaxMC and PVL, popular UED score functions, in terms of their predictiveness of learnability. However, there is no demonstration of a similar analysis for SFL. Although SFL's scoring function is intuitive and it is not as challenging to guess what the scoring would look like, including such an illustrative comparison would support the claims in this paper.

- The empirical results according to the proposed evaluation protocol for robustness (Figures 3a, 5a, and 7a) show that SFL outperforms the rest of the evaluated methods for alpha < 1 (100%). However, except in the multi-agent case (Fig 5a), the expected success rates indicate marginal or no improvement. In MiniGrid, the difference between DR and SFL drops quickly as alpha gets above 0.1 (10%). This is likely due to SFL not doing so well in easier levels compared to other baselines. Although this is expected, as the proposed metric results in upsampling learnable environments, not easy ones, I suggest a comparison of results in easy levels to conclude whether this is the case or not.

**Questions:**

Table 4 showcases compute times for each evaluated approach. Most of these approaches have similar components, except maybe SFL's extra rollout phase. So why is SFL one of the fastest?

**Limitations:**

- The major limitation of the proposed approach is the rollouts needed to compute the learnability score of randomly sampled levels. Section 4.2 indicates that 5000 levels are sampled, and n-step rollouts, where n is not specified, are generated to compute the scores. It would be more informative if the amount of computation and time spent in this phase were reported (in comparison to the usual curriculum learning task sampling phase). In addition, I wonder how SFL is one of the fastest in the single agent nav environment, as reported in Table 4 in Appendix F, despite having this additional phase.

---

> ### Author Rebuttal · Authors · 2024-08-06
>
> We thank the reviewer for their comments, especially for highlighting our "simple yet novel UED approach" and that our paper is "well-written".
> Below, we address the issues raised in the review:
>
> # Correlation analysis for SFL
>
> >Section 4.1 analyses MaxMC and PVL, popular UED score functions, in terms of their predictiveness of learnability. However, there is no demonstration of a similar analysis for SFL. Although SFL's scoring function is intuitive and it is not as challenging to guess what the scoring would look like, including such an illustrative comparison would support the claims in this paper.
>
>
> We agree such analysis would be helpful, and we have added this to our updated manuscript. Due to space constraints we have not included this in the attached rebuttal PDF.
>
>
> # Easy Levels
>
> To assess performance on easy levels we have run our evaluation procedure over 10,000 uniformly sampled levels with fewer obstacles than usual. For JaxNav, we used a maximum fill % of $\leq 30\%$, half of the standard 60%. Meanwhile, for Minigrid, we use a maximum number of 30 walls instead of 60. These levels, therefore, are generally easier than the levels we evaluated on in the main paper. Results are reported in Figure 4(a,b) of the attached PDF.
>
> On JaxNav, SFL still demonstrates a significant performance increase while on Minigrid all methods are very similar. Due to the challenging dynamics of JaxNav, even levels with a small number of obstacles can present difficult control and navigation problems meaning Automated Curriculum Learning (ACL) methods (such as SFL) still lead to a performance differential over DR. Meanwhile, in Minigrid, due to its deterministic dynamics, difficulty is heavily linked to the obstacle count as this allows for more complex mazes. As such, DR is competitive to ACL methods in settings with fewer obstacles.
>
> # SFL runtime speed
>
> Thank you for raising this important point, as it highlights how SFL takes advantage of parallel computations on the GPU.
>
> Looking at a single iteration (including training + eval) in Minigrid on an L40S GPU, the time breakdowns are as follows:
> |                      | PLR   | SFL   |
> | -------------------- | ----- | ----- |
> | Train Step           | 37.5s | 35s   |
> | Get Learnable Levels | 0     | 2.2s  |
> | Eval Step            | 0.7s  | 0.7s  |
> | Total                | 38.2s | 37.9s |
>
> We note that the SFL rollouts are fast for two reasons:
> - We aggressively parallelise them, running up to 5000 environments in parallel, which takes about the same time as running only hundreds in parallel.
> - We do not compute any gradients for these transitions.
>
> The upshot is that they take significantly less time than the actual training step. Furthermore, UED's training step is more complex than SFL's, since it must maintain a buffer of levels, compute the scores during training, and potentially update the buffer.
>
> For JaxNav, obtaining the learnable levels takes a little longer, but still takes less time than our logging. Due to this, slight differences in logging between PLR and SFL masked this additional computational cost. To better control for this, we measure how long SFL takes vs PLR in the absence of logging. We find that SFL is about 6% slower than PLR, despite running all of the additional rollouts. In Figure 1(a) of the attached PDF (and our updated manuscript), **we have provided compute-time matched results** for single-agent JaxNav (where SFL runs for 6% fewer timesteps). Despite running for fewer timesteps, SFL still significantly outperforms UED methods. SFL on Minigrid runs as fast, or slightly faster than PLR.
>
> We apologise for the oversight in not specifying how long our rollouts are. We use a consistent rollout length of 2000 steps. We have updated this in our manuscript.
>
> # Conclusion
> We hope we have addressed the reviewer's comments and are happy to discuss any of these points further. We would further ask if our responses have addressed the reviewer's concerns, and that they consider increasing their support for our paper.

---

> > ### Comment · Reviewer_gAKk · 2024-08-12
> > **Re: Rebuttal by Authors**
> >
> > Thank you for your effort in responding to my comments and questions and providing new results that support the validity of your work.
> >
> >  My concerns have been thoroughly addressed, so I'll raise my score from 6 to 7.

---

### Official Review · Reviewer_Su34 · 2024-07-14

**Soundness:** 3
**Presentation:** 3
**Contribution:** 3
**Rating:** 7
**Confidence:** 5

**Summary:**

The paper introduces a new metric based on solve rates for evaluating the learning potential of tasks in a multi-task RL environment. It also presents a new evaluation protocol based on worst-case performance to better characterize the robustness of different methods. The work uses this protocol to compare their method with several Unsupervised Environment Design baselines and concludes that it outperforms the baseline. They also find that their UED baselines perform worse than Domain Randomization on this new evaluation metric, and conduct additional experiments to confirm and analyze these results.

**Strengths:**

The paper includes many contributions, including a new method, a new evaluation protocol, and new benchmarks of existing methods. These contributions are open sourced and well documented in the appendix to support reproducibility. The work attempts to identify failure modes of a large class of regret-based UED algorithms which could help to move further ACL research toward more promising directions. These methods are effective and widely applicable, but their generalization outside of a small set of baselines has not been thoroughly studied. This work will become increasingly important as RL research moves on to more complex environments.

**Weaknesses:**

The main weakness of the paper is the claim that domain randomization and their new heuristic outperform all UED algorithms, which is not entirely convincing from the results. The PLR hyperparameters used in this work are different from [1] and [2], for example the prioritization, temperature, and number of edits are all different. PLR in particular has several hyperparameters that need to be tuned for new domains, such as the buffer size, sampling temperature, and staleness coefficient. The authors do not explain whether they tuned the hyperparameters for the baselines or how they tuned the hyperparameters for their own method. I think even a small grid search over reasonable parameters that worked in other domains would help to make the comparisons throughout this paper more convincing.

Another concern is that this work does not include any standard episodic return plots, which is the main metric of comparison in RL. Without these plots, it is difficult to tell whether their method truly outperforms the baselines, or whether their baselines are properly tuned. For instance, if the SFL algorithm performs better according to the CVaR evaluation but not on a standard test return plot, then it's debatable which method is more useful. That being said, [2] does compare the mean evaluation solve rate of UED baselines on Minigrid, and finds that Accel and PLR both significantly outperform DR. In this work, PLR severely underperforms DR using the same metric in Figure 3. This seems to be strong evidence that its hyperparameters or implementation are not correct.

Even if we assume that the baselines are working as intended, I PLR and SFL have many differences which makes it unclear what change is actually leading to better performance. I think this work would benefit from additional experiments using the learnability metric in SFL as the prioritization metric in PLR, and possibly the alternative, using PVL and MaxMC as the selection metrics in the full SFL algorithm.

Overall the writing and presentation was quite good, but there were a few notable issues. The paper somewhat confounds its description of PLR with Robust PLR in section 2.2.1. PLR trains on randomly sampled levels, while Robust PLR evaluates randomly sampled levels and only trains on levels from its replay buffer. Also, "learnability" is not clearly defined until 4.2, after it has been referenced many times. This is ok when discussing "intuitive notions of learnability" in a vague sense, but not when making statements such as "no correlation exists between learnability and regret score".

[1] Jiang, Minqi, et al. "Replay-guided adversarial environment design." Advances in Neural Information Processing Systems 34 (2021): 1884-1897.
[2] Parker-Holder, Jack, et al. "Evolving curricula with regret-based environment design." International Conference on Machine Learning. PMLR, 2022.

**Questions:**

* Why did you not simply use your heuristic instead of PVL or MaxMC in prioritized level replay?
* Why does there appear to be a discrepancy between the UED baseline solve rates reported in this paper and the ones reported in [2]?
* On line 214 you describe sampling 10,000 solvable levels to evaluate on. Are these unseen levels (at least with high probability) or are they sampled from the training set?
* You mention in the Limitations section that your method is restricted to Jax-based settings. Is the method limited to those settings, or just your particular Jax-based implementation?

**Limitations:**

The authors adequately discuss the limitations in the paper and throughout the work. Their SFL method appears to be less general that the UED baselines that they compare against, but performs much better on the domains tested in this work. Simple methods on new benchmarks can serve as stepping stones to more general methods, so I do not believe this impacts their contribution.

---

> ### Author Rebuttal · Authors · 2024-08-06
>
> Dear reviewer, thank you for your thorough review and helpful suggestions! We also appreciate you mentioning that our paper "includes many contributions" and "could help to move further ACL research toward more promising directions." Please find our responses and changes below.
>
> # Hyperparameters and UED underperformance
>
> As rightfully requested by the reviewer, we have performed extensive hyperparameter tuning for all methods (excluding DR). Using these new, optimised hyperparameters (see Figure 1 in the attached 1-page PDF), **UED outperforms DR** (in Minigrid and Single-agent Jaxnav) but **SFL still outperforms UED**. For DR, we use the PPO hyperparameters as discussed in the main response, and did not tune it further, as it has no other relevant hyperparameters. Please see our main response for details regarding how we performed our sweep.
>
> Furthermore, when comparing our results to previous work please note that in the original Robust PLR paper, they use 25 walls for Minigrid. However, they also show that changing the number of walls can drastically alter results; for instance, in appendix C.1.2, they show that using 50 walls causes DR to be competitive to UED. Following Minimax [1] and JaxUED [2], we use 60 walls. Therefore, we cannot directly compare the results of our work and that of the original Robust PLR & ACCEL papers.
>
> [1] Jiang, et al. "Minimax: Efficient Baselines for Autocurricula in JAX."
>
> [2] Coward, et al. "JaxUED: A simple and useable UED library in Jax."
>
> # Episodic Return Plots
>
> We primarily include success on the set of held-out levels during training rather than episodic return because, in these goal-oriented domains, the success rate is a more intuitive and clearer marker of performance. This is because the actual environment reward may include reward shaping terms that do not directly correspond to our ultimate aim of reaching the goal (but are nonetheless necessary to learn a policy).
>
> We also find that the average return in all our domains is closely tied to success rate. This is illustrated in Figure 5 of the manuscript where 5\(c) reports success rate while 5(d) reports return (note, there is a naming error on the y-axis of Figure 5(d)).
>
> **Therefore, the performance trends illustrated in 3(b), 5\(c), 7(b) are mirrored in the episodic return throughout training.** We have clarified this in the revised manuscript, and have included reward curves in the appendix.
>
> # Learnability as PLR score function and PVL as SFL's metric
>
> Thank you for suggesting this, we agree that this is an important ablation to understand the source of improvement and have included these results in the attached PDF (Figure 3).
>
> We ran this ablation for both Minigrid and single-agent JaxNav, and in both domains, **SFL + Learnability outperforms all other combinations**. Using learnability with PLR/ACCEL has a slight positive effect for PLR in JaxNav and a negative effect in Minigrid. Using PVL as the score function inside our SFL algorithm performs worse, except when we have a large buffer of levels in JaxNav.
>
> Some intuition for why this is:
> - In the large buffer case for PVL on JaxNav, the training levels are more random, since we also include levels with lower scores. Indeed, with a small buffer, PVL performs very poorly, a result in line with our analysis in Figure 2 of the submitted manuscript as PVL fails to predict the frontier of learnability accurately.
> - Conversely, a large buffer results in a set of levels that is slightly biased towards higher success rates, as illustrated in Figure 2 of the submitted manuscript (i.e., PVL correlates slightly with success rate). This bias helps screen out some unsolvable levels using the scoring function, a feature of regret-based UED algorithms [3]. Additionally, our observation shows that random JaxNav levels often provide a good learning signal, as demonstrated by a sampling ratio $\rho$ of 0.5 performing similarly to 1.0 in the SFL ablations shown in Figure 2 of the 1-page PDF. Therefore, SFL-PVL with a large batch of levels creates a learning curriculum that leads to decent performance, although it is still inferior to SFL with learnability.
>
>
> **The results of this ablation suggest that the improvements due to SFL can be attributed to both the learnability score function *and* our improved sampling approach.**
>
> Note, however, that we did not hyperparameter-tune these results, as we used the optimised hyperparameters found above, and just swapped the score function.
>
> [3] Jiang, et al. "Replay-guided adversarial environment design."
>
> # Poor explanations and presentation
>
> Thank you for pointing out these problems, we appreciate your thoroughness. We have rectified them in our updated manuscript. We have also defined learnability earlier on, making the paper clearer.
>
> # Questions
> > Using our heuristic with PLR?
>
> See above and Figure 3 in the attached 1-page PDF. We find that SFL + Learnability outperforms the other combinations.
>
> > A discrepancy between UED's performance in this paper and [2]?
>
> See above, in "Hyperparameters and UED underperformance".
>
> > Are the 10,000 solvable levels unseen?
>
> The levels used during evaluation are sampled from our environment generator separately from the training process. As the space of possible levels is very large, generating the same random level twice is unlikely.
>
> > Is the method restricted to Jax-based settings?
>
> Our implementation is in JAX but the method is general. However, as raised in our limitations section, one must take the cost of SFL's additional environment rollouts into account when considering implementing our algorithm; we chose JAX because its speed and parallelisation significantly alleviates this constraint.
>
> # Conclusion
> We hope that the reviewer feels we have addressed their questions and welcome any further discussion. We also ask that, if all their concerns are met, the reviewer consider revising their score to recommend accepting the paper.

---

> > ### Comment · Reviewer_Su34 · 2024-08-13
> >
> > I’d like to thank the authors for their very thorough response and comprehensive additional experiments, as well as apologize for not responding sooner.
> >
> > Thank you for performing a more thorough hyperparameter sweep, and for pointing out the difference in minigrid block budgets from prior work. After looking through JaxUED and Robust PLR again, I agree with the authors that UED is comparable to DR in the 50 and 60 block setting. That being said, I think the authors should also experiment in the more challenging 25 block setting where UED methods are most effective. Regardless, these new results provide convincing evidence that the proposed method is better than the SOTA in at least some settings.
> >
> > Than you also for running the requested ablations so quickly. The results seem inconsistent across hyperparameters and environments, but it does appear that SFL is generally better, and both the sampling and prioritization metrics positively impact performance.
> >
> > My main concern was that there seemed to be strong evidence that the evaluations in the paper were not fair. With these extensive new results and changes, I believe the contribution of this work is convincing, and I hope the authors will incorporate these new results in their paper as promised. I will raise my score to a 7.

---

### Author Rebuttal · Authors · 2024-08-06

Dear reviewers, we appreciate your detailed reviews and concrete suggestions for improvement. We are especially grateful for reviewers mentioning that our paper is "well-written", "technically sound", and "includes many contributions", including an algorithm that is "novel, simple, and easy to understand." We are pleased that reviewers recognise the "simulation support" for our claims and that all "contributions are open-sourced and well documented in the appendix to support reproducibility".

We have implemented your suggested improvements, and run additional experiments, which **confirm our findings that SFL outperforms current UED methods on several domains.**

In our global response, we highlight the main changes we made, with updated and additional results presented in the 1-page PDF of figures.

# New Environment - XLand-Minigrid
*Requested by Reviewer ycoo.*

We have added an additional environment to our experiments by running DR, SFL and PLR on Xland-Minigrid's meta-RL task [1]. See below for details.

**Results**
We report performance using our CVaR evaluation procedure and, in line with [1], as the mean return on an evaluation set during training. Our results are presented in Figures 1(d) and 4(d) of the attached PDF. **SFL outperforms both PLR and DR.** Results are averaged over 5 seeds and during evaluation each ruleset was rolled out for 10 episodes. Training runs were conducted on one L40S and due to the large number of levels being rolled out to fill SFL's buffer, SFL was slower than DR and PLR. As such, we report results for SFL compute-time matched to PLR. Not only does SFL outperform PLR for their respective best set of hyperparameters but it is also much more robust to hyperparameter choice, with only one configuration of PLR's hyperparameters being competitive compared to the large majority of SFL's. More details on this experiment are in our updated manuscript.


**Environment Overview**
This domain combines an XLand-inspired system of extensible rules and goals with a Minigrid-inspired goal-oriented grid world to create a domain with a diverse distribution of tasks. Each task is specified by a ruleset, which combines rules for environment interactions with a goal, and [1] provide a database of presampled rulesets for use during training. Following [1], we use a 13x13 grid with 4 rooms and sample rulesets from their high diversity benchmark with 3 million unique tasks. As training involves sampling from a database of precomputed rulesets, ACCEL is not applicable. PLR and SFL select rulesets for each meta-RL step to maximise return on a held-out set of evaluation rulesets.


**Hyperparameter tuning**
All methods used the default PPO hyperparameters given by [1]. For both PLR and SFL, we performed a grid search using similar compute budgets. Due to space, we cannot provide details here but have in our updated manuscript.

[1] Nikulin, et al. "XLand-minigrid: Scalable meta-reinforcement learning environments in JAX."

# Extensive Hyperparameter Tuning
*Suggested by Reviewers Su34 and ycoo.*

To ensure competitive results, we have performed extensive hyperparameter tuning for each baseline. Updated results are shown in Figure 1 of the 1-page PDF. **We find that SFL still significantly outperforms existing UED methods.**

For PPO we conducted an extensive sweep for the JaxNav environment ensuring robust DR performance. For Minigrid, our JaxNav PPO parameters performed similarly to those given in the JaxUED implementation but allowed us to use 256 environment rollouts in parallel during training compared to JaxUED's 32. We kept these base PPO parameters fixed for all methods.

For both PLR and ACCEL, we swept hyperparameters for Minigrid and JaxNav's single-agent variation. For both methods, we performed a grid search over the replay $p$, in $\{0.5, 0.8\}$, level buffer size $K$, in $\{1000, 4000, 8000\}$, choice of scoring function in $\{ \text{PVL}, \text{MaxMC}\}$ and prioritisation function in $\{ \text{rank}, \text{TopK}\}$. For rank prioritisation, we searched for the temperature in $\{0.3, 1.0\}$, while for TopK we searched for k in $\{1, 32, 128\}$. For ACCEL, we additionally searched for the number of edits in $\{5, 20, 50\}$. Each set of hyperparameters tested was run over 3 seeds. For multi-agent JaxNav, we used the best-performing hyperparameters for single-agent JaxNav.

For SFL, we started with the default parameters listed in the main text and conducted an independent line search over the parameters: batch size $N \in \{ 500, 5000, 25000\}$, rollout length $L \in \{1000, 2000, 4000 \}$, number of levels to save $K \in \{100, 1000, 5000 \}$, buffer update period $T \in \{10, 50, 500, 1000, 2000 \}$ and sampled environments ratio $\rho \in \{0.25, 0.5, 0.75, 1.0\}$.
Since this is a line search and not a grid search, the total number of tuning runs (and total compute) is significantly less than for PLR and ACCEL (20 runs for SFL vs 90 for PLR & 270 for ACCEL).

# Additional Baselines
*Suggested by Reviewer Su34.*

We have added two new baselines (see Figure 3 in the 1-page PDF):
1. Positive Value Loss (PVL) as a score function for SFL.
2. Learnability as a score function for PLR and ACCEL.

**SFL+Learnability still outperforms both of these variations.**

# Ablations and Effects of SFL Hyperparameters
*Suggested by Reviewer ycoo.*

In Figure 2 of the 1-page PDF, we have investigated the effects of changing hyperparameter values on SFL. We also experimented with different variations of learnability, where the highest learnability score corresponds to success probabilities other than 0.5, for instance, where solving a level 20% of the time is defined as "optimal" learnability (Figure 4\(c)).

Overall, SFL's performance improves when we sample more levels, and it is relatively robust to all other hyperparameters. There is a detrimental effect if we sample the set of learnable levels too infrequently (see the "Buffer Update Period" subfigure).

---

### Decision · Program_Chairs · 2024-09-25

**Decision:**

Accept (poster)

**Comment:**

This paper introduces a novel metric for evaluating the learning potential of tasks in a multi-task RL environment. It also proposes a new evaluation protocol to better characterize the robustness of different methods, and compares its proposed method with several Unsupervised Environment Design baselines using this protocol. This paper shows that its proposed method outperforms the baselines.

This paper is very interesting and I believe that many researchers in the field of RL will find this paper important. All the reviewers recommend to accept this paper, and three reviewers have given a rating of 7. After reading the paper, the reviews, and the rebuttals, I agree with the reviewers and recommend to accept this paper.